# Stem-loop and circle-loop TADs generated by directional pairing of boundary elements have distinct physical and regulatory properties

**Wenfan Ke[1†], Miki Fujioka[2†], Paul Schedl[1]\*, James B Jaynes[2]\***

[1]Department of Molecular Biology, Princeton University, Princeton, United States;
[2]Department of Biochemistry and Molecular Biology, Thomas Jefferson University, Philadelphia, United States

**\*For correspondence:**
pschedl@princeton.edu (PS);
james.jaynes@jefferson.edu (JBJ)

[†]These authors contributed equally to this work

**Competing interest:** The authors declare that no competing interests exist.

**Abstract** The chromosomes in multicellular eukaryotes are organized into a series of topologically independent loops called TADs. In flies, TADs are formed by physical interactions between neighboring boundaries. Fly boundaries exhibit distinct partner preferences, and pairing interactions between boundaries are typically orientation-dependent. Pairing can be head-to-tail or head-to-head. The former generates a stem-loop TAD, while the latter gives a circle-loop TAD. The TAD that encompasses the *Drosophila even skipped* (*eve*) gene is formed by the head-to-tail pairing of the *nhomie* and *homie* boundaries. To explore the relationship between loop topology and the physical and regulatory landscape, we flanked the *nhomie* boundary region with two attP sites. The attP sites were then used to generate four boundary replacements: *λ* DNA, *nhomie forward* (WT orientation), *nhomie reverse* (opposite of WT orientation), and *homie forward* (same orientation as WT *homie*). The *nhomie forward* replacement restores the WT physical and regulatory landscape: in MicroC experiments, the *eve* TAD is a 'volcano' triangle topped by a plume, and the *eve* gene and its regulatory elements are sequestered from interactions with neighbors. The *λ* DNA replacement lacks boundary function: the endpoint of the 'new' *eve* TAD on the *nhomie* side is ill-defined, and *eve* stripe enhancers activate a nearby gene, *eIF3j*. While *nhomie reverse* and *homie forward* restore the *eve* TAD, the topology is a circle-loop, and this changes the local physical and regulatory landscape. In MicroC experiments, the *eve* TAD interacts with its neighbors, and the plume at the top of the *eve* triangle peak is converted to a pair of 'clouds' of contacts with the next-door TADs. Consistent with the loss of isolation afforded by the stem-loop topology, the *eve* enhancers weakly activate genes in the neighboring TADs. Conversely, *eve* function is partially disrupted.

## eLife assessment

This **valuable** work investigates the role of boundary elements in the formation of 3D genome architecture. The authors established a specific model system that allowed them to manipulate boundary elements and examine the resulting genome topology. The work yielded the first demonstration of the existence of stem and circle loops in a genome and confirms a model which had been posited based on extensive prior genetic work, providing insights into how 3D genome topologies affect enhancer–promoter communication. The evidence is **solid**, although the degree of generalization remains uncertain.

## Introduction

Chromosomes in multicellular animals are organized into a series of topologically independent looped domains, called TADs, or topologically associating domains (*Cavalheiro et al., 2021*; *Chetverina et al., 2017*; *Jerković et al., 2020*; *Matthews and White, 2019*; *Rowley and Corces, 2018*). The arrangement of TADs in a given chromosomal DNA segment is generally (though not precisely) similar in different tissues and developmental stages, and this is a reflection of the mechanism underlying TAD formation—the endpoints of TADs are determined by a special class of elements called chromatin boundaries or insulators. While boundary-like elements have been identified in a wide range of animals and plants, the properties of this class of DNA elements have been most fully characterized in *Drosophila* (*Cavalheiro et al., 2021*; *Chetverina et al., 2017*). Fly boundaries have one or more large (100–400 bp) nucleosome-free nuclease-hypersensitive sequences that are targets for multiple DNA binding chromosomal architectural proteins. While only a single chromosomal architectural protein, CTCF, has been characterized in mammals, there are several dozen such proteins in flies, and the list is still growing (*Heger et al., 2013*; *Heger and Wiehe, 2014*; *Schoborg and Labrador, 2010*). In addition to subdividing the chromosome into a series of loops, fly boundary elements have insulating activity. When placed between enhancers or silencers and their target promoters, boundaries block regulatory interactions (*Bell et al., 2001*; *Chetverina et al., 2014*; *Chetverina et al., 2017*). This activity provides a mechanism for delimiting units of independent gene activity: genes located between a pair of compatible boundaries are subject to regulatory interactions with enhancers/silencers present in the same chromosomal interval, while they are insulated from the effects of enhancers/silencers located beyond either boundary in adjacent regulatory neighborhoods. It is currently thought that their ability to organize the chromosome into topologically independent loops is important for their insulating activity (*Cai and Shen, 2001*; *Gohl et al., 2011*; *Muravyova et al., 2001*).

Studies dating back to the 1990s have suggested that fly boundaries subdivide the chromosome into loops by physically pairing with each other (*Chetverina et al., 2014*; *Chetverina et al., 2017*). In these first experiments, regulatory interactions were observed for transgenes inserted at distant sites in chromosome that were carrying either the *gypsy* transposon boundary *su(Hw)* or the bithorax complex (BX-C) boundary *Mcp* (*Muller et al., 1999*; *Sigrist and Pirrotta, 1997*; *Vazquez et al., 1993*). Further support for the idea that boundaries function by pairing has come from chromatin immunoprecipitation, chromosome conformation capture (CCC), MicroC, and direct imaging experiments (*Chen et al., 2018*; *Li et al., 2011*; *Vazquez et al., 2006*). More recent studies have revealed physical interactions in the CNS, such as those found for *su(Hw)* and *Mcp,* which 'reach over' multiple intervening TADs, consistent with them playing an important role in cell type-specific gene regulation (*Mohana et al., 2023*) by bringing distant enhancers and promoters together.

The parameters governing pairing interactions have been defined using insulator bypass, transvection, and boundary competition assays. These studies have shown that fly boundaries are able to pair not only with heterologous boundaries but also with copies of themselves. Moreover, the pairing interactions typically exhibit a number of characteristic features: promiscuity coupled with clear partner preferences, and orientation dependence.

Partner preferences depend upon the chromosomal architectural proteins that interact with each boundary. For example, in the boundary bypass assay, a set of enhancers are placed upstream of two reporters (*Cai and Shen, 2001*; *Kyrchanova et al., 2008a*; *Muravyova et al., 2001*). When multimerized dCTCF sites are placed between the enhancers and the closest reporters, both reporters are insulated from the enhancers. When a second set of multimerized dCTCF sites are placed downstream of the closest reporter, bypass is observed. In this case the closest reporter, which is bracketed by the multimerized dCTCF sites, is still insulated from the enhancers; however, the downstream reporter is activated (*Muravyova et al., 2001*). Heterologous combinations give a different result: when multimerized dCTCF sites are placed upstream of the closest reporter and multimerized Zw5 sites are placed downstream, no bypass is observed. Endogenous fly boundaries also show partner preferences in bypass assays and in boundary competition experiments (*Gohl et al., 2011*; *Kyrchanova et al., 2011*; *Kyrchanova et al., 2008b*). On the other hand, while boundaries have partner preferences, they are also promiscuous in their ability to establish functional interactions with other boundaries. For example, the *Fab-8* insulator can partner with *scs'* from the *Drosophila* heat shock locus (*Gohl et al., 2011*).

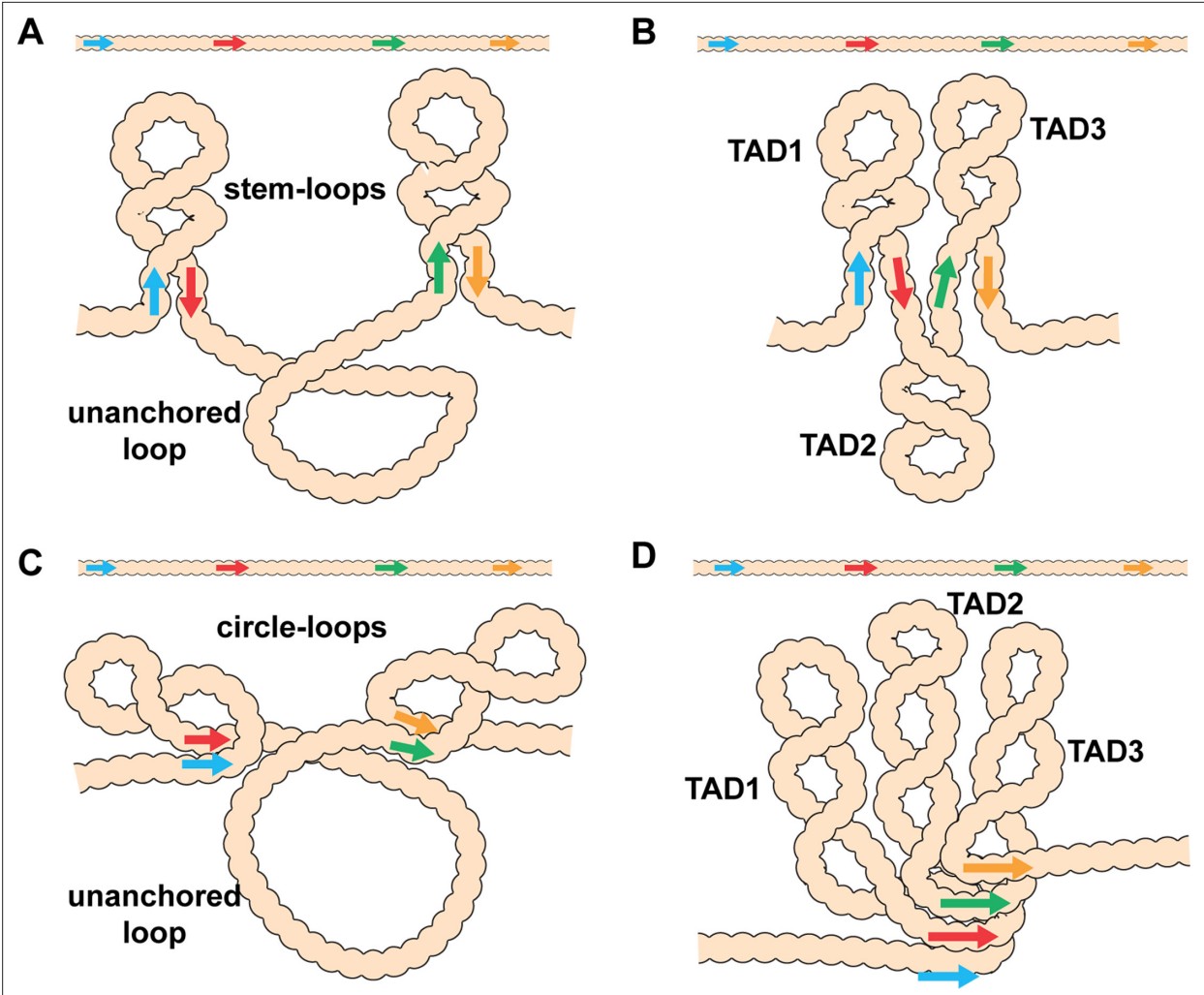

**Figure 1.** Diagram of the possible loop topologies generated by head-to-head and head-to-tail pairing. (**A**) Head-to-tail boundary pairing (arrows) generates a series of stem-loops linked together by an unanchored loop. In this case, the main axis of the chromosome would correspond to the unanchored loops connecting different stem-loops. (**B**) If boundaries pair with both neighbors (head-to-tail), the stem-loops would be linked to each other by the paired boundaries. In this case the main axis of the chromosome would correspond to the paired boundaries. (**C**) Head-to-head boundary pairing generates a series of circle-loops linked together by an unanchored loop. The unanchored loop will be the main axis of the chromosome. (**D**) If boundaries pair with both neighbors (head-to-head), the chromatin fiber will be organized into a series of circle-loops connected to each other at their base, and these paired boundaries will define the chromosomal axis. In both (**B**) and (**D**), the pairing interactions between the blue and red boundaries need not be in register with the pairing of the red boundary to the next-door green boundary. In this case, the main axis of the chromosome may bend and twist, and this could impact the relative orientation of the stem-loops/circle-loops. More complex structures would be generated by mixtures of stem-loops and circle-loops.

In addition, to partner preferences, pairing interactions between endogenous fly boundaries are, with a few exceptions, orientation-dependent. Self-pairing interactions are head-to-head. This seems to be a common feature of fly boundaries and has been observed for *scs*, *scs'*, *iA2*, *wari*, *Mcp*, *Fab-8*, *AB-I*, *homie*, and *nhomie* (*Fujioka et al., 2016*; *Kyrchanova et al., 2008a*). In contrast, pairing interactions between heterologous boundaries can be head-to-head or head-to-tail. The two boundaries bracketing the *even-skipped* (*eve*) locus, *homie* and *nhomie*, pair with each other head-to-tail, while boundaries in the *Abdominal-B* (*Abd-B*) region of the BX-C usually pair with their neighbors head-to-head. The topology of the loops (TADs) generated by head-to-tail and head-to-head pairing in *cis* between neighboring boundaries is distinct. As illustrated in *Figure 1*, head-to-tail pairing generates stem-loops, while head-to-head pairing generates circle-loops. The loops could be connected to each other by unanchored loops (*Figure 1A and C*), or they could be linked directly to each other if boundaries can pair simultaneously with both neighbors (*Figure 1B and D*). An alternating pattern of TADs

connected by DNA segments that crosslink to each other with reduced frequency (c.f., $\lambda$ *DNA* below) is not often observed in MicroC experiments. Instead, most TADs appeared to be directly connected to both of their neighbors without an intervening unanchored loop (*Batut et al., 2022*; *Bing et al., 2024*; *Levo et al., 2022*; see also below). This would suggest that TAD boundaries are typically linked to both neighbors, either simultaneously or as alternating pair-wise interactions.

Key to understanding the 3D organization of chromosomes in multicellular eukaryotes will be the identification of TADs that are stem-loops and TADs that are circle-loops. In the studies reported here, we have used MicroC to analyze the contact maps generated by stem-loops and circle-loops. Stem-loop and circle-loop TADs are expected to interact differently with their neighbors, and this should be reflected in the patterns of crosslinking events between neighboring TADs. As illustrated for linked stem-loops in *Figure 1B*, TAD2 is isolated from its next-door neighbors, TAD1 and TAD3. In this configuration, crosslinking events between sequences in TAD2 and sequences in TAD1 and TAD3 will be suppressed. On the other hand, TAD1 and TAD3 are in comparatively close proximity, and crosslinking between sequences in these two TADs is expected to be enhanced. A different pattern of neighborly interactions is expected for circle-loop TADs. In this case, the TAD in the middle, TAD2, is expected to interact with both of its neighbors (*Figure 1D*). To test these predictions, we have first compared the MicroC contact profiles for stem-loop and circle-loop TADs. For stem-loops we selected the *eve* TAD, while for circle-loops we chose the four TADs that comprise the *Abd-B* parasegment-specific regulatory domains. We show that these stem-loop and circle-loop TADs have distinctive crosslinking signatures. To confirm these MicroC signatures, we converted the topology of the *eve* TAD from a stem-loop to a circle-loop. In addition to changing the MicroC signature of the *eve* TAD, the change in topology is accompanied by changes in the regulatory interactions between *eve* and its neighbors.

## Results

### Stem-loops versus circle-loops

The distinctive loop topologies of stem-loops and circle-loops are expected to be reflected in the contact maps that are generated in MicroC experiments. To determine if this is the case, we compared the MicroC contact maps for the *eve* TAD and the TADs that correspond to the four *Abd-B* parasegment-specific regulatory domains, *iab-5*, *iab-6*, *iab-7,* and *iab-8*. The *eve* TAD is generated by pairing interactions between the *nhomie* boundary upstream of the *eve* transcription unit and the *homie* boundary downstream. Since *nhomie* and *homie* pair with each other head-to-tail, the *eve* TAD has a stem-loop topology (*Fujioka et al., 2016*). Unlike the *eve* boundaries, the boundaries that delimit the *Abd-B* regulatory domains are thought to pair with their neighbors head-to-head (*Chetverina et al., 2017*; *Kyrchanova et al., 2008a*; *Kyrchanova et al., 2011*; *Kyrchanova et al., 2008b*). This means that the parasegment-specific regulatory domain TADs, *iab-5*, *iab-6*, *iab-7,* and *iab-8*, are expected to have a circle-loop topology.

As shown in *Figure 2*, the *eve* TAD and the four *Abd-B* TADs have distinctive MicroC contact patterns. The *eve* TAD is a 'volcano' triangle with a plume. The endpoints of the volcano triangle are delimited by *nhomie* on the left and *homie* on the right, and within the *eve* locus (the volcano), there are additional enhanced interactions. While the volcano triangle is generated by contacts between sequences within the *eve* stem-loop, contacts between sequences in *eve* and in the neighboring TAD on the left, TL (which contains multiple sub-TADs and six genes: *CG15863, CG1418, Pal1, CG12133, eIF3j,* and *CG12134*), are much reduced (L-ev in *Figure 2A*). There is a similar suppression of contacts between sequences in the *eve* TAD and sequences in the large neighboring TAD on the right, TM (which contains *TER94* and *Pka-R2*; ev-M in *Figure 2A*). On the other hand, as expected from the regulatory interactions observed for stem-loops in boundary bypass experiments (*Kyrchanova et al., 2008a*), physical contacts between sequences in TL and TM are enhanced compared to those between *eve* and TL (L-ev) or TM (ev-M). Because of the preferential interactions between TADs to either side of the *eve* stem-loop, the *eve* volcano triangle is topped by a plume (L-M in *Figure 2A*). TM also interacts with the two TADs farther to the left of *eve*, TK (K-M in *Figure 2A*), and TJ (J-M in *Figure 2A*; see also *Figure 2—figure supplement 1*).

Like the *eve* boundaries, the TADs in the *Abd-B* region of BX-C region are connected to their neighbors by the boundaries at their base. As predicted from genetic studies on BX-C boundaries,

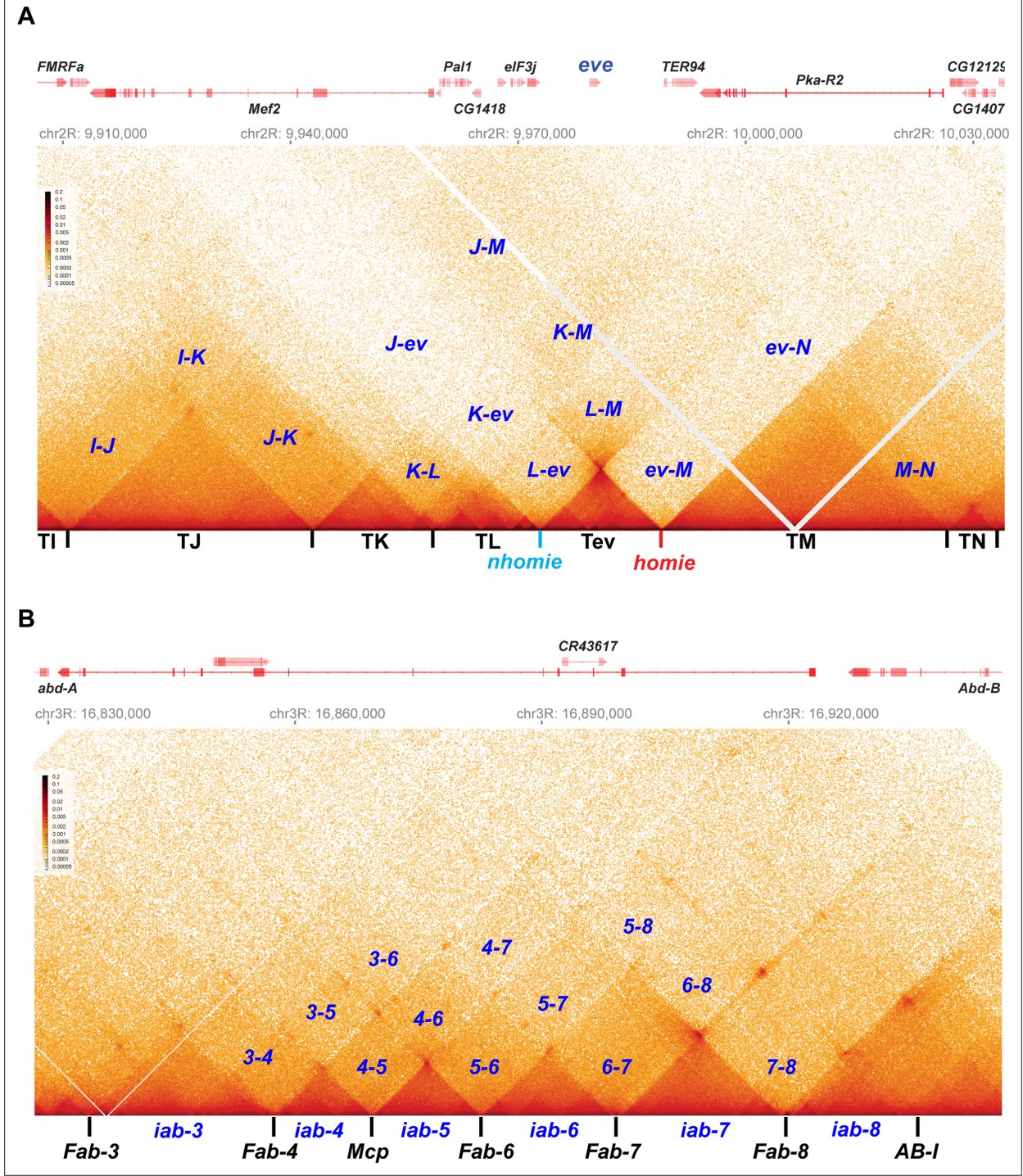

**Figure 2.** Stem-loops and circle-loops. "*Once in a while you get shown the light in the strangest of places if you look at it right.*"[a] MicroC contact profile for *Drosophila* wild-type (*yw*) NC14 embryos. The bin size for each panel is 200 bp. (**A**) *eve* and neighboring TADs (TI, TJ, TK TL, TM, and TN). The *eve* TAD is a volcano with a plume that is anchored by *nhomie* (*nh*) and *homie* (*h*). The plume is generated by crosslinking of sequences in the two neighboring TADs. At the bottom of the plume, TL sequences are linked to sequences in TM close to *eve*, including *TER94*. At the next level, sequences in TK are linked to TM (region K-M). In addition, sequences in TL are linked to sequences in TM located beyond the *TER94* gene. At the next level, sequences in TJ are linked to sequences in TM. Note that interactions between sequences in TL and TJ and sequences in TM close to the *eve* TAD are somewhat less frequent than those farther away from the *eve* TAD. Sequences in the neighboring TADs also interact with each other, as indicated. For example, sequences in TK and TJ interact with each other (J–K) and also interact with sequences in TI (I–K and I–J). (**B**) The BX-C gene *Abd-B* and the parasegment- (PS-) specific regulatory domains *iab-3, iab-4, iab-5, iab-6, iab-7,* and *iab-8*. *iab-4* regulates the *abd-A* gene in PS9, while *iab-5 – iab-8*

*Figure 2 continued on next page*

*Figure 2 continued*

regulate *Abd-B* in PSs 10–13, respectively. These domains are separated from each other by the boundary elements *Fab-4*, *Mcp*, *Fab-6*, *Fab-7*, and *Fab-8*, as indicated. The *AB-I* boundary is located upstream of the *Abd-B* promoter. Each regulatory domain corresponds to a TAD. Though partially insulated from each other, each TAD interacts with its immediate neighbors. For example, *iab-5* interacts with its immediate neighbors *iab-4* and *iab-6* to give *4–5* and *5–6*, respectively. It also interacts with the next-next-door neighbor *iab-7* (*5-7*) and even its next-next-next-door neighbor *iab-8* (*5-8*). (ᵃ From 'Scarlet Begonias' by the Grateful Dead, 1974).

The online version of this article includes the following figure supplement(s) for figure 2:

**Figure supplement 1.** MicroC contact profiles for *nhomie forward*, *lambda DNA*, *nhomie reverse*, and *homie forward* in larger scale.

each TAD corresponds to one of the four parasegment-specific regulatory domains. While the MicroC contact maps for the four *Abd-B* TADs resemble the contact patterns in the *eve* TAD, these *Abd-B* TADs differ from *eve* in that there are no plumes above their triangle peaks (**Figure 2B**). Instead, the *Abd-B* TADs are overlaid by a series of rectangular interlocking low-density contact (LDC) domains, or clouds. As illustrated in **Figure 2B**, the *iab-6* regulatory domain is flanked by clouds generated by crosslinking with its next-door neighbors *iab-5* (5–6) and *iab-7* (6–7), followed by crosslinking with neighbors that are a TAD away from *iab-6*, *iab-4* (4–6), and *iab-8* (6–8). The other regulatory domains also form a unique set of interlocking LDCs/clouds with their immediate neighbors, their next-next-door neighbors, and their next-next-next-door neighbors.

## TAD formation in a *nhomie* deletion

To further investigate the pairing properties and functioning of the *eve* boundaries, we used CRISPR-Cas9 to add two attP sites flanking the *nhomie* region, replacing the region with a mini-*white* gene. Using mini-*white* as an exchange marker, ΦC31 recombinase-mediated cassette exchange (RMCE) was used to restore the sequence of the region, with *nhomie* modifications. As a control for possible effects of the sequences introduced in generating the modification, we reinserted a 597 bp *nhomie* DNA fragment in the same orientation as the endogenous *nhomie* boundary (*nhomie forward*). To maintain roughly the same distance between *eve* and the neighboring TAD in the *nhomie* deletion, we introduced a 606 bp DNA fragment from phage $\lambda$ ($\lambda$ DNA). **Figure 3** (and **Figure 2—figure supplement 1A and B**) shows the MicroC contact profiles for the *nhomie forward* and $\lambda$ DNA replacements in 12–16 hr embryos (mid-embryogenesis: stages 12–14). Except that the sequencing depth of the *nhomie forward* replacement is not as great as the WT shown for the *eve* locus in **Figure 2**, the profile is quite similar. Like WT, there are sub-TADs within the *eve* TAD. One of these appears to link *homie* and the neighboring PRE to the *eve* promoter-proximal PRE (**Fujioka et al., 2008**), and is marked by an interaction dot (asterisk in **Figure 3A**). Another links *nhomie* to the *eve* promoter region (blue arrow in **Figure 3A**). The *eve* TAD is topped by a plume, which is generated by interactions between sequences in neighboring TADs TL with TM (L-M). On the other hand, interactions between *eve* and its neighbors are suppressed. Like *eve*, there are sub-TADs in the neighboring TAD, TL. The TL sub-TAD closest to *eve* (TL4) corresponds to the *CG12134* transcription unit (green arrowhead marks the boundary: **Figure 3**), while the neighboring sub-TAD (TL3) encompasses the *eIF3j* transcription unit (blue arrowhead: **Figure 3**).

The MicroC profile of the $\lambda$ DNA replacement (**Figure 3D**) is quite different from that of either *nhomie forward* or WT, which are similar (**Figures 2A and 3A**). While *homie* still defines the distal (relative to the centromere) end of the *eve* locus, the $\lambda$ DNA replacement does not function as a TAD boundary, and the leftward endpoint of the *eve* TAD is no longer well-defined. One new 'endpoint' for the *eve* locus maps to sequences between *CG12134* and *eIF3j* (green arrowhead), which in wild type corresponds to the left boundary of the TL sub-TAD TL4. The other endpoint maps to sequences between *eIF3j* and *CG12133* (blue arrowhead), which in wild type define the left boundary of the TL sub-TAD TL3. These interactions are not as stable as those between *nhomie* and *homie* as the density of interaction dots is lower. Furthermore, they appear to flip back and forth between alternative endpoints (as indicated by the green and blue double arrows in **Figure 3F**) based on the MicroC contact profile, which is consistent with a mixture of (at least) two conformations. The *eve* TAD also interacts with sequences in the two other TL sub-TADs, TL1 and TL2. In addition, the *eve* promoter appears to interact with sequences located upstream of *CG12134* (purple arrowhead in **Figure 3D** and double arrow in **Figure 3F**), while this interaction is not observed in the *nhomie forward* replacement. While the TL TAD (from the TK:TL boundary to *nhomie*) is also disrupted by the *nhomie* deletion

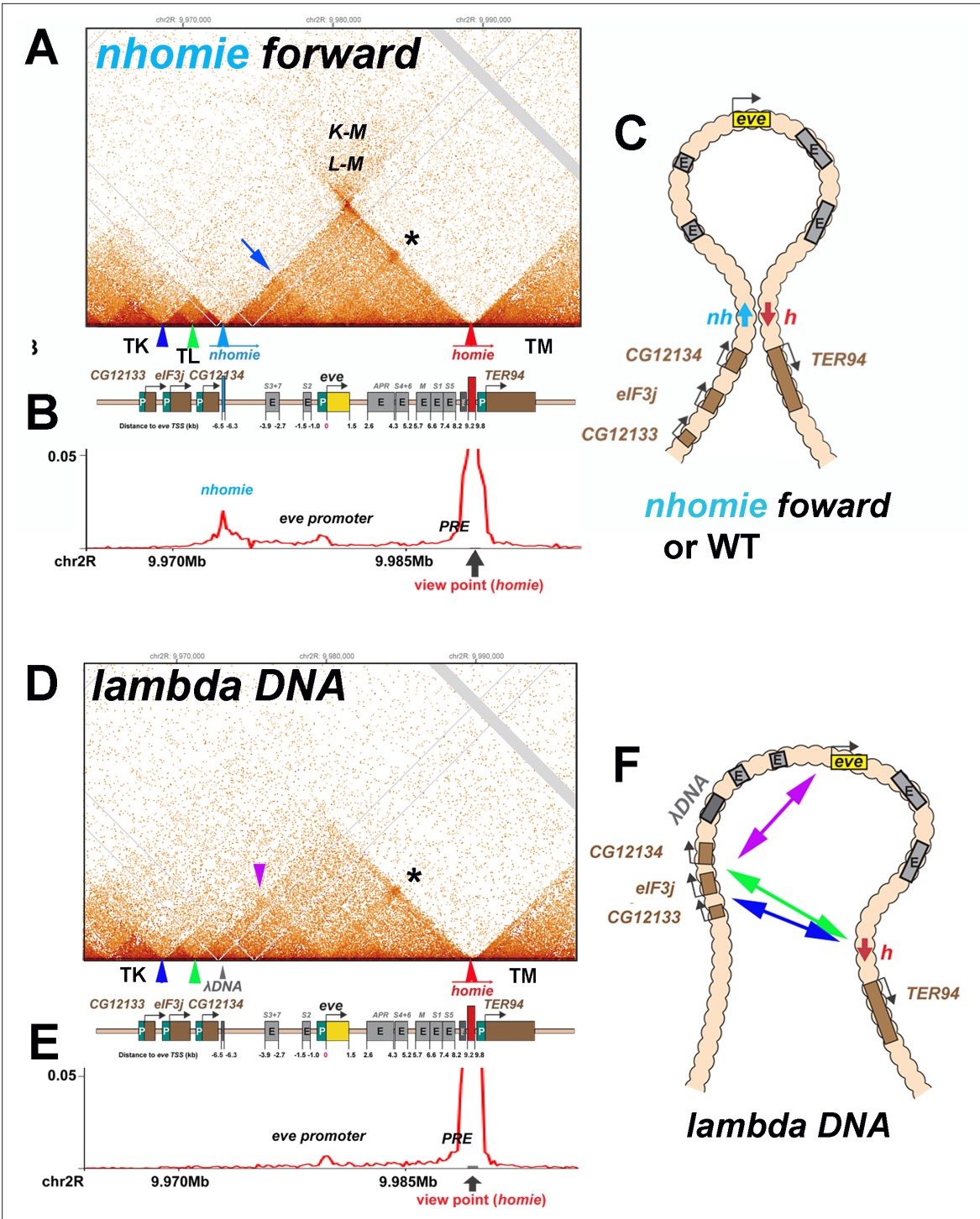

**Figure 3.** TAD organization of the *nhomie forward* and *lambda DNA* replacements. (**A**) MicroC contact profile of 12–16 hr embryos (stage 12–14) *nhomie forward* embryos. In this, our positive control, *nhomie* replaces endogenous *nhomie*, in the same orientation. N (replicates) = 2. Resolution = 200 bp. L-M: interactions between TADs TL and TM flanking the *eve* locus. Asterisk: sub-TAD linking the *eve* promoter to the *eve* PRE and *homie*. Dark blue arrow: sub-TAD linking the *eve* promoter to *nhomie*. Light blue arrow: *nhomie*. Red arrow: *homie*. Green arrowhead: sub-TAD boundary formed by the *CG12134* promoter region. Dark blue arrowhead: sub-TAD boundary formed by *eIF3j* promoter region. Diagram: map of *eve* locus and surrounding genes. (**B**) Virtual 4C with viewpoint from *homie* (black arrow) in *nhomie forward* embryos. (**C**) Diagram of the *eve* stem-loop TAD. (**D**) MicroC contact profile of 12–16 hr *λ DNA* embryos. In this replacement, *λ* DNA is inserted in place of *nhomie*. N (replicates) = 3. Resolution = 200 bp. Asterisk: sub-TAD linking the *eve* promoter to the *eve* PRE and *homie*. Purple arrowhead: sub-TAD linking *CG12134* promoter region to the *eve* promoter. The *eIF3j* sub-TAD TL3 (between the blue and green arrowheads) is still present. (**E**) Virtual 4C with viewpoint from *homie* (black arrow) in *λ DNA* embryos. (**F**) Diagram of the 'unanchored' *eve* TAD. Double arrows show novel interactions.

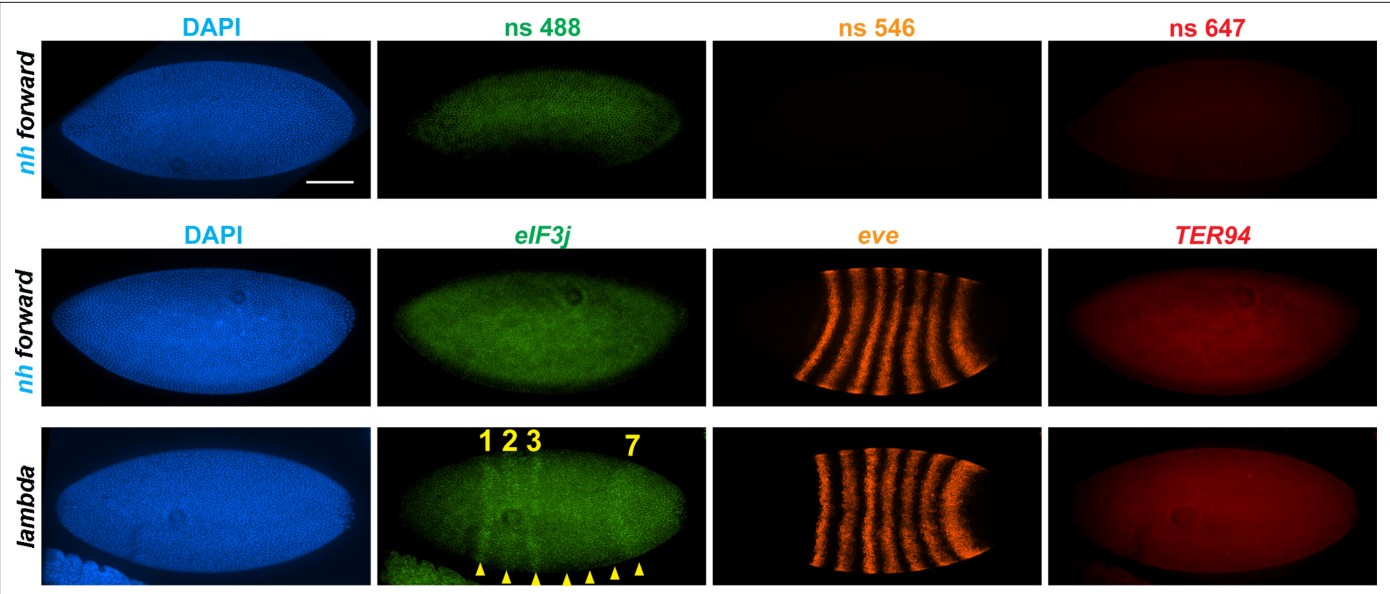

**Figure 4.** *nhomie* deletion (*λ DNA* replacement) exposes *eIF3j* to the *eve* enhancers. *nh forward*: positive control, as in **Figure 3**. *λ DNA: nhomie* is replaced with *λ* DNA. At the syncytial blastoderm stage, a series of stripe-specific enhancers upstream (stripes 1, 2, 3, 7) and downstream (stripes 1, 4, 5, 6) of the *eve* gene drive *eve* expression. During cellularization of the blastoderm and gastrulation, a single enhancer located upstream of *eve* drives expression of all seven stripes. DAPI: DNA stained with DAPI (blue). *eIF3j*: embryo hybridized with probe complementary to *eIF3j* mRNA. *eve*: embryo hybridized with probe complementary to *eve* mRNA. *TER94*: embryo hybridized with probe complementary to *TER94*. Yellow arrowheads: *eve*-enhancer-driven *eIF3j* stripes. Control nonspecific probes for each channel indicate autofluorescence background in the top panel. Scale bar = 100 μm.

The online version of this article includes the following figure supplement(s) for figure 4:

**Figure supplement 1.** Expression of *CG12134* in WT (*yw*) and the four *nhomie* replacements.

**Figure supplement 2.** Expression of *eIF3j* (*Adam*) and *TER94* in WT (*yw*) and the four *nhomie* replacements.

(it has a much less distinct 'volcano apex', and its right-most sub-TAD TL4 is now fused with the *eve* TAD), the left-most TL sub-TADs (TL1, TL2, and TL3) are still present, indicating that their formation does not depend on *nhomie*. As shown in the virtual 4C at *homie* viewpoint in **Figure 3B and E**, the *homie* boundary interacts with the *nhomie forward* replacement, but does not contact the *λ DNA* replacement.

### *eve* enhancers activate *eIF3j* expression in the *nhomie* deletion

In transgene assays, boundary elements block regulatory interactions when interposed been enhancers (or silencers) and reporter genes (**Chetverina et al., 2014**; **Chetverina et al., 2017**; **Kellum and Schedl, 1992**). To determine whether this is also true in their endogenous context, we compared the expression in syncytial blastoderm embryos of the two genes that flank *nhomie* at the *eve* locus, *eIF3j* and *CG12134*. As the *nhomie* deletion eliminates *homie*'s pairing partner and disrupts the *eve* TAD, we also examined the expression of *eve* and of the gene just beyond the *homie* boundary, *TER94*, which has strong maternal expression through stage 11 (**Figure 4—figure supplement 2B**, WT). Consistent with the seemingly normal MicroC profile on the *homie* side of the *eve* TAD, we did not detect evidence of *eve*-like *TER94* expression (**Figures 4 and 5**). Thus, the formation and functioning of the TM TAD do not appear to be impacted by either the loss of *nhomie per se* or the fact that the left end of the *eve* TAD is no longer properly anchored.

In the case of the gene closest to the *nhomie* deletion, *CG12134*, we were unable to consistently detect transcription driven by the *eve* enhancers in either *nhomie forward* or *λ DNA* embryos. In some *λ DNA* embryos, there were hints of stripes at the blastoderm stage (see **Figure 4—figure supplement 1**); however, these 'stripes' were not observed in most embryos. Since *CG12134* (which forms the TL4 sub-TAD in wild type) is closest to the *eve* enhancers and interacts most strongly, it is possible that the promoter is not compatible with the *eve* enhancers. A different result was obtained for *eIF3j* in *λ DNA* embryos. As shown in the HCR-FISH experiment in **Figure 4** and quantitated in **Figure 5**,

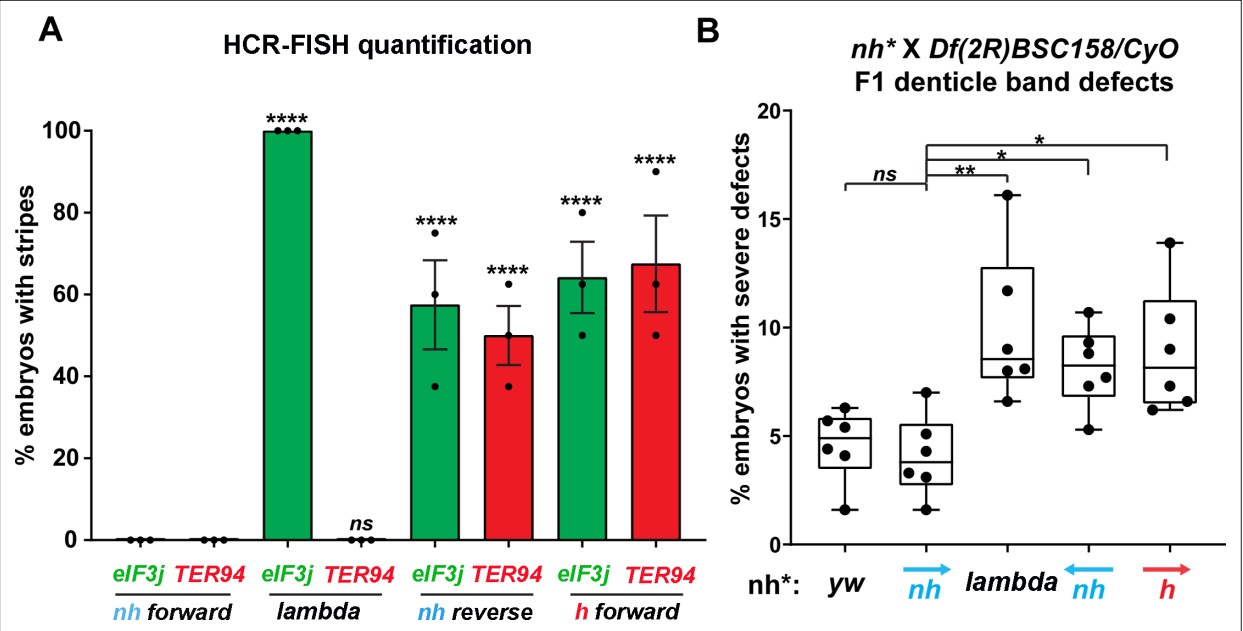

**Figure 5.** Manipulating the *nhomie* boundary impacts the regulatory landscape. N = # of independent replicates, n = # of embryos. Two-way ANOVA with Tukey's multiple comparisons test for each pair of groups was used to determine the statistical significance. *p≤0.05, **p≤0.01, ***p≤0.001, and ****p≤0.0001. (**A**) Quantitation of the number of embryos showing stripe patterns in HCR-FISH for *eIF3j* and *TER94,* as shown in *Figures 4 and 7*. N = 3. n = 45 for each group. (**B**) Quantitation of the number of missing ventral denticle bands in larvae from a cross of *BSC/CyO,hb-lacZ* deficiency females to males of the indicated genotypes (N = 6): wild-type control (*yw*), n = 767. For the *nhomie forward* replacement, n = 1099; for the *λ DNA* replacement, n = 1175; for the *nhomie reverse* replacement, n = 1083; for the *homie forward* replacement, n = 1137.

we observed a series of *eIF3j* stripes over a dark background in pre-cellular blastoderm embryos. As this background hybridization is evident in earlier stages, much of it is likely to be of maternal origin. In contrast to the *λ DNA* replacement, these stripes are not visible in the *nhomie forward* (control) replacement (*Figures 4 and 5A*). While it is possible to detect all seven stripes in *λ DNA* blastoderm stage embryos, their levels of expression are not equal. The highest levels correspond to *eve* stripes 1, 2, 3, and 7, while *eve* stripes 4, 5, and 6 are expressed at much lower levels. Since the stripe enhancers for 1, 2, 3, and 7 are located between the *eve* promoter and *nhomie*, they are closer to the *eIF3j* promoter than the enhancers for stripes 4, 5, and 6, which are located downstream of the *eve* transcription unit. In addition to possible effects of distance, the subdomain linking the *eve* PRE and *homie* to the promoter is still observed in the *λ DNA* replacement, and this could partially sequester the enhancers located downstream of the *eve* promoter.

We also used digoxigenin *in situ* hybridization to analyze *eIF3j* expression. With this procedure we were able to detect a low level of *eve*-activated *eIF3j* stripe expression in WT and *nhomie forward* embryos at stage 5 (blastoderm) and stages 7–8 (early gastrula: *Figure 4—figure supplement 2A*). In stage 5 embryos when *eve* expression is driven by specific stripe enhancers, *eIF3j* expression appears to be similar in all seven stripes (*Figure 4—figure supplement 2*, WT and *nhomie forward*). In contrast, as was observed in the HCR-FISH experiments (*Figure 4*), there is a clear bias for enhancers located upstream of *eve,* where *eIF3j* is also located, in the *λ DNA* replacement at this point in development. In stage 7–8 embryos, the seven-stripe enhancer drives *eve* expression. It is located close to *nhomie* and, not surprisingly, high levels of *eIF3j* expression are observed in all seven stripes in the *λ DNA* replacement (*Figure 4—figure supplement 2A*). At later embryonic stages, *eve* expression is driven by tissue-specific neurogenic, mesodermal, and anal plate enhancers. However, *eIF3j* is expressed at high levels in a complex pattern in older embryos, and we were unable to unambiguously detect expression driven by the *eve* enhancers over this background mRNA. This is also not surprising, given that all of the enhancers for these aspects of *eve* expression are located downstream of the *eve* promoter, like the enhancers driving stage 5 stripes 4, 5, and 6.

While the *nhomie* deletion did not have any obvious impact on the level or pattern of *eve* expression in blastoderm stage embryos (see *Figure 4*), it seemed possible that *eve* activity was not entirely

normal. To test this possibility, we mated males homozygous for either *λ DNA* or *nhomie forward* to females heterozygous for a chromosomal deficiency that includes the *eve* gene. We then quantitated the number of missing denticle bands in embryonic cuticle preps. As shown in *Figure 5B*, the frequency of larvae with 'severe' defects (two or more missing ventral denticle bands) in *nhomie forward* is similar to that in a WT *yw* control. In contrast, in the *λ DNA* replacement, the frequency of larval cuticles with two or more missing denticle bands is increased nearly twofold. Taken together, the increase in severity of the cuticle defects is significant at the $p<0.01$ level (one-tailed *t*-test). The A6 denticle band is missing most frequently, followed by A2, A4, and then A8. These even-numbered abdominal denticle bands are those that are lost in *eve* deficiency mutants (from which the name *even skipped* comes), suggesting that *eve* stripe expression at blastoderm stages is compromised in the embryos that produce these defective cuticles.

## The *eve* TAD is converted from a stem-loop to a circle-loop by inverting *nhomie*

The orientation of boundary:boundary pairing interactions determines the topology of each chromatin loop (*Bing et al., 2024*; *Fujioka et al., 2016*). Since *nhomie* and *homie* pair with each other head-to-tail, the endogenous *eve* TAD is a stem-loop. This orientation dependence means that one can convert the *eve* TAD from a stem-loop to a circle-loop by inverting the *nhomie* boundary. If our expectations are correct, the MicroC contact pattern will also be transformed from a volcano triangle with a plume to one in which sequences in the *eve* TAD are flanked by a cloud of crosslinked sequences from both neighboring TADs (TL and TM), like that observed in the *Abd-B* region of the BX-C.

We tested this prediction by inserting the *nhomie* boundary in the reverse orientation (*nhomie reverse*). *Figure 6A* shows that the *eve* TAD is reconstituted by *nhomie reverse* (compare with *Figure 3*: see also *Figure 2—figure supplement 1*). The sub-TAD evident in the *nhomie forward* replacement linking *nhomie* to the *eve* promoter is also re-established (blue arrow). In addition, consistent with our expectation, the plume topping the *eve* TAD is gone and is replaced by a much more sparsely populated LDC domain (purple double-arrow and above). The more prominent LDC TAD-TAD interactions (the clouds) are between sequences in the *eve* TAD and the neighboring TADs. On the right, *eve* forms an LDC interaction domain with TM (ev-M). On the left, *eve* interacts most strongly with sequences in TL4, and progressively less strongly with sequences in the sub-TADs TL3, TL2, and then TL1 (L-ev). In addition to restoring the *eve* TAD, the TL TAD is re-established, indicating that the *nhomie* boundary is important in defining both endpoints of the TL TAD. On the other hand, with the exception of the *CG12133* sub-TAD, *nhomie* does not play a role in generating the three other sub-TADs in the TL TAD. The other interesting feature is a 45° band of interaction (just below the purple double-arrow) that includes interactions between sequences in TL4 and sequences in *eve* that appear to be located near the left edge of *homie*. These sequences likely correspond to the *eve* 3′ PRE (*Fujioka et al., 2008*), located just inside the 3′ end of the *eve* TAD.

## Insulation is reduced in *nhomie reverse*

Consistent with the models for stem-loops and circle-loops in *Figure 1*, the neighborly interactions evident in the MicroC contact patterns for *nhomie forward* and *nhomie reverse* are quite distinct. The *nhomie forward* TAD is isolated from its neighbors, and crosslinking between *eve* and the neighboring TADs is suppressed (*Figures 1B and 2A*). This is not true for the circle-loop TAD generated by *nhomie reverse*: in this configuration, the *eve* TAD is not sequestered from neighboring TADs (*Figures 1D and 2B*), but instead interacts much more frequently with sequences in next-door TADs than in the stem-loop configuration. Since the *eve* TAD is no longer as well-isolated from its neighbors, this could increase the frequency of 'productive' interactions between *eve* regulatory elements and genes in nearby TADs, and vice versa.

To test whether the circle-loop topology has an impact on the regulatory landscape, we hybridized *nhomie reverse* embryos with HCR-FISH probes for *eIF3j*, *TER94,* and *eve*. In early blastoderm stage *nhomie forward* embryos, there is little evidence of *eIF3j* or *TER94* expression driven by *eve* stripe enhancers, and the HCR-FISH hybridization pattern is uniform (see *Figure 4*). In contrast, it is possible to discern individual stripes of both *eIF3j* and *TER94* mRNA over the background signal in a subset of *nhomie reverse* blastoderm stage embryos in HCR-FISH (*Figure 7*). Since these genes are assembled into their own topologically independent looped domains rather than being in the same domain as

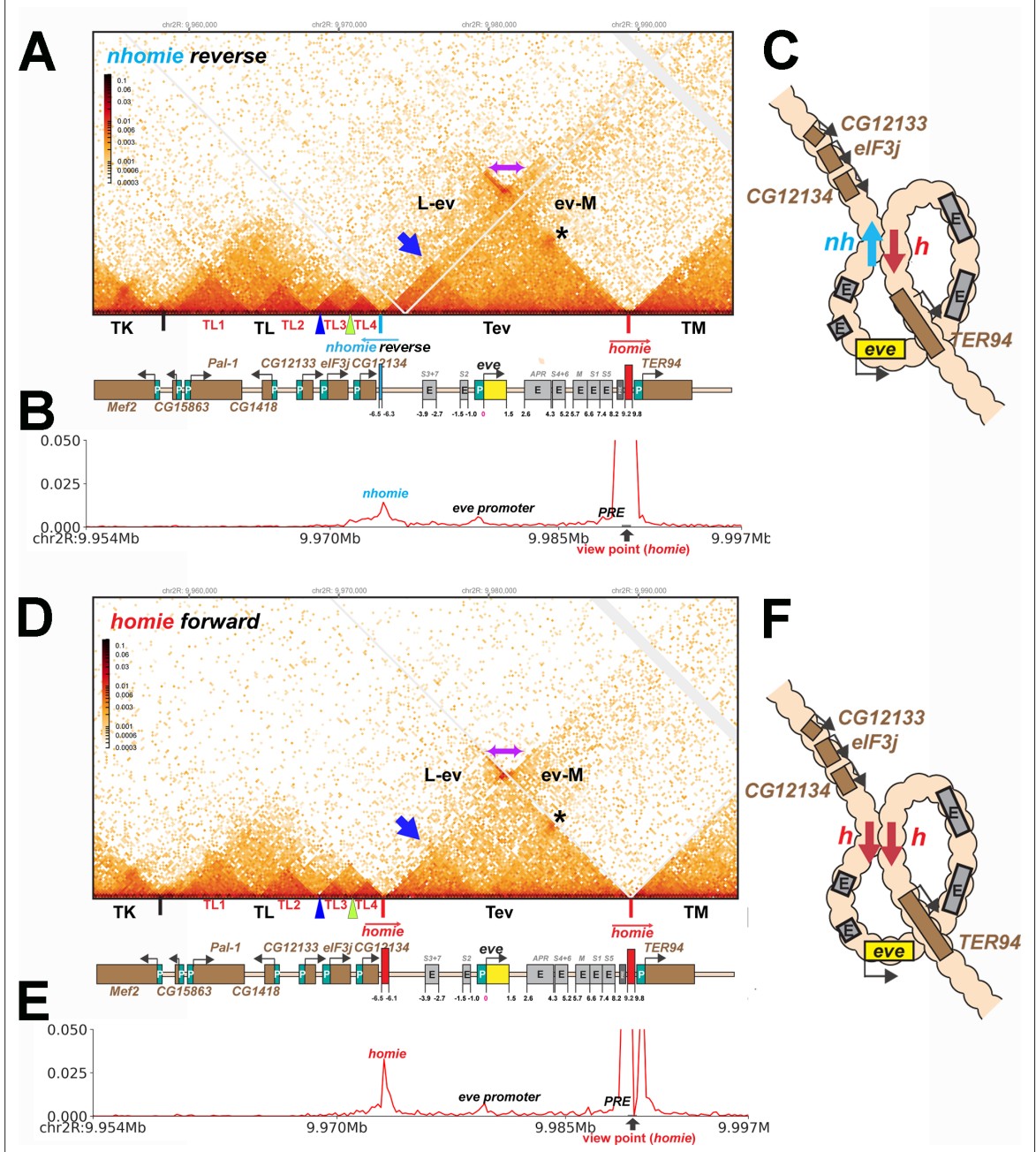

**Figure 6.** TAD organization of the *nhomie reverse* and *homie forward* replacements. (**A**) MicroC contact profile of 12–16 hr *nhomie reverse* embryos. In this replacement, *nhomie* is inserted in the reverse orientation compared to WT *nhomie*. N (replicates) = 3. Resolution = 200 bp. (**B**) Virtual 4C with viewpoint from *homie* (black arrow) in *nhomie reverse* embryos. (**C**) Diagram of the *nhomie reverse:homie* circle-loop. (**D**) MicroC contact profile of 12–16 hr *homie forward* embryos. In this replacement, *homie* is inserted in the forward orientation (the same as the endogenous *homie*): N (replicates) = 3, resolution = 200 bp. (**E**) Virtual 4C with viewpoint from *homie* (black arrow) in *homie forward* embryos. (**F**) Diagram of the *homie forward:homie* circle-loop. (**A, C**) Note that interactions between the TADs flanking the *eve* locus (purple double arrow) are suppressed compared to *nhomie forward* (see *Figure 3*), while interactions of the *eve* TAD (Tev) with TL and TM are enhanced (L-ev and ev-M). Asterisk: sub-TAD linking the *eve* promoter to the *eve* PRE and *homie*. Dark blue arrow: sub-TAD linking the *eve* promoter to *nhomie reverse*. Light blue arrow: *nhomie reverse*. Red arrow: *homie*. Green arrowhead: sub-TAD boundary formed by the *CG12134* promoter region. Dark blue arrowhead: sub-TAD boundary formed by the *eIF3j* promoter region.

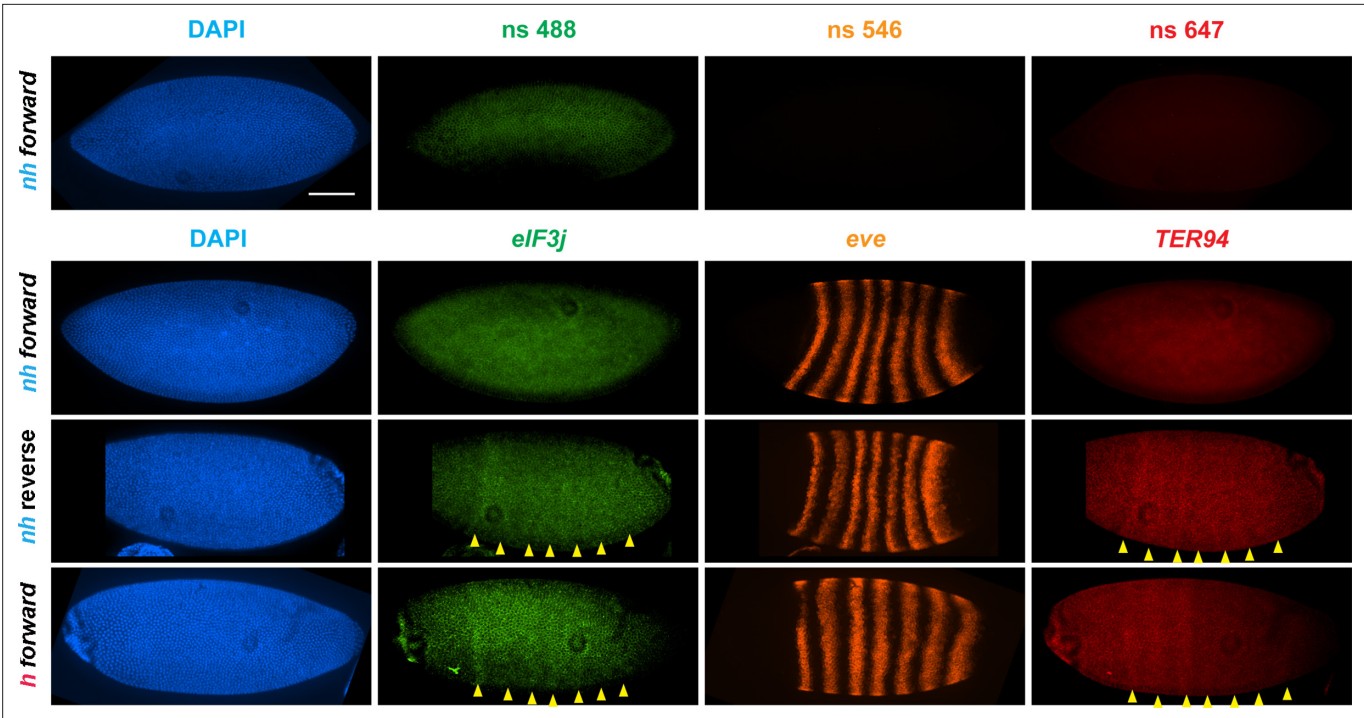

**Figure 7.** *eve* enhancers activate neighboring genes when the *eve* TAD is a circle-loop. HCR-FISH hybridization to mRNA expressed by *eIF3j*, *eve* and *TER94* at the blastoderm stage (embryonic stage 5). *nh forward*: *nhomie* is replaced with *nhomie* in the forward (normal) orientation (positive control, as in **Figure 3**). *nh reverse*: *nhomie* is replaced with *nhomie* in the reverse orientation. *h forward*: *nhomie* is replaced with *homie* in the forward orientation. Yellow arrowheads: positions of stripes. DAPI (blue): DNA stained with DAPI. *eIF3j* (green): embryo is hybridized with probe complementary to *eIF3j* mRNA. *eve* (orange): embryo is hybridized with probe complementary to *eve* mRNA. *TER94* (red): embryo is hybridized with probe complementary to *TER94*. Control nonspecific probes for each channel indicate autofluorescence background in the top panel. Scale bar = 100 μm.

*eve*, the level of *eIF3j* and also *TER94* stripe expression is lower than that observed in the *nhomie* deletion (λ DNA). The *nhomie reverse* circle-loop also differs from the *nhomie* deletion (λ DNA) in that there is not such an obvious preference for which *eve* enhancers activate expression. In addition, *eve*-dependent *eIF3j* and *TER94* stripes are detected in only about half of the blastoderm stage embryos (**Figure 5A**). It is possible that the frequency of productive inter-TAD contacts differs from one embryo to the next; however, a more likely reason is that the high background of *eIF3j* and *TER94* transcripts obscures the low level of *eve* enhancer-driven expression at this stage. Once the seven-stripe enhancer is activated, *eve*-dependent *TER94* expression in *nhomie reverse* is elevated, and all seven stripes are observed (**Figure 4—figure supplement 2B**). This fits with the MicroC contact profile. As shown in **Figure 6A**, the *TER94* gene is preferentially crosslinked to *eve* sequences located between *nhomie reverse* and the *eve* promoter compared to sequences spanning the *eve* gene and the downstream enhancers (i.e., the upper-left portion of the ev-M region shows more crosslinking than does the lower-right portion). This bias correlates with the location of the seven-stripe enhancer, located near the 5′ end of the *eve* TAD. By contrast, there is much less seven-stripe enhancer-driven expression of *eIF3j* (**Figure 4—figure supplement 2A**) than of *TER94*. Consistent with this observation, crosslinking between *eIF3j* and sequences in *eve* close to *nhomie reverse* and the seven-stripe enhancer occur less frequently than crosslinking to sequences on the other side of the *eve* TAD (i.e., the lower-left portion of the L-ev region of the MicroC profile shows less crosslinking than does the upper-right portion), although this crosslinking bias may not be as pronounced as that observed between *TER94* and the two sides of the *eve* TAD.

While *eve* stripe expression is not discernibly different from wild type, the circle-loop topology still impacts *eve* function. As shown in **Figure 5B**, the fraction of *nhomie reverse* embryos with two or more missing denticle bands is nearly twice that in either *yw* or the *nhomie forward* replacement. Taken together, the increase in severity of the cuticle defects is significant at the p<0.05 level (one-tailed

*t*-test). As was observed for the *λ DNA* replacement, the A6 denticle band is missing most frequently, followed by A2, A4, and then A8.

### *homie forward* converts the *eve* stem-loop into a circle-loop

While the findings in the previous section show that loop topology impacts how sequences in TADs interact with each other and with their neighbors, one might argue that the effects we observed are a reflection of some novel properties of the *nhomie* boundary when it is inverted. To test this possibility, we took advantage of the fact that in addition to pairing with *nhomie*, the *homie* boundary pairs with itself (*Fujioka et al., 2016*). However, unlike *nhomie:homie* pairing, which is head-to-tail, *homie:homie* pairing is head-to-head. This means that it is possible to convert the *eve* TAD into a circle-loop by inserting *homie* into the *nhomie* deletion in the forward orientation.

As shown in *Figure 6D* (*Figure 2—figure supplement 1*), the MicroC contact profile of *homie forward* is similar to that observed for *nhomie reverse*. The plume topping the *eve* TAD in wild type (*Figure 2A*) or in *nhomie forward* (*Figure 3A*) is absent. Likewise, instead of being isolated from its neighbors, the *eve* TAD contacts TL and TM. Also like *nhomie reverse*, *homie forward* forms a subdomain within the *eve* TAD linking it to sequences in the *eve* promoter. There is also enhanced cross-linking between sequences in TL4 and sequences on the right end of the *eve* TAD that correspond to the *eve* 3′ PRE (*Fujioka et al., 2016*).

The MicroC pattern is not the only similarity between *homie forward* and *nhomie reverse*. The functional properties of the *homie forward eve* TAD are also similar. *Figure 7* shows that the *eve* enhancers weakly activate both *eIF3j* and *TER94*. As was the case for *nhomie reverse*, expression levels at the blastoderm stage are low and are observed in only about half of the embryos (*Figure 5A*). After the blastoderm stage, when the seven-stripe enhancer drives *eve* expression, an even higher level of *TER94* expression is observed (*Figure 4—figure supplement 2B*). In addition, the functioning of the *eve* gene when it is in the circle-loop configuration is not as efficient, and the frequency of *homie forward* embryos with two or more missing denticle bands is twice that of *nhomie forward* (*Figure 5B*). Taken together, the increase in severity of the cuticle defects is significant at the p<0.05 level (one-tailed *t*-test).

## Discussion

Two different though overlapping classes of chromosomal architectural elements have been identified in flies. One class is the PREs found in many developmental loci. PREs were first discovered because they induce pairing-sensitive silencing of reporter genes (*Americo et al., 2002*; *Kassis et al., 1991*). More recent studies have shown that the ability of these elements to physically pair with each other may be their most important function (*Batut et al., 2022*; *Levo et al., 2022*). The other class of architectural elements are chromatin boundaries (also called insulators). PRE pairing in *cis* typically takes place within the context of a larger chromosomal domain, or TAD. In contrast, boundary elements are responsible for defining the endpoints of these looped domains (*Arzate-Mejía et al., 2020*; *Batut et al., 2022*; *Bing et al., 2024*; *Chetverina et al., 2017*; *Ibragimov et al., 2023*; *Stadler et al., 2017*). While not much is known about the parameters governing PRE pairing, the pairing interactions of fly boundaries have been studied in some detail. The key features include an ability to engage in promiscuous pairing interactions, distinct partner preferences, and orientation dependence. Of the endogenous (non-*gypsy*) boundaries whose functional properties have been studied in detail, only one, *Fab-7*, appears to be able to pair in both orientations. However, *Fab-7* may be unusual in that its boundary activity depends upon factors that have been implicated in the functioning of PREs (*Kyrchanova et al., 2018*). For all of the other boundaries studied so far, pairing interactions are orientation-dependent. When fly boundaries pair with themselves, the interactions are head-to-head (*Kyrchanova et al., 2008a*). This make sense, as the available evidence suggests that self-pairing interactions in *trans* may be largely responsible for the pairing of homologous chromosomes in precise register (*Erokhin et al., 2021*; *Fujioka et al., 2016*). In this case, head-to-tail self-pairing would uncouple the loops on the two homologs (and sister chromatids).

Unlike self-pairing in *trans*, pairing interactions between heterologous boundaries in *cis* can be head-to-head or head-to-tail. The topological consequences are quite distinct. The former generates a circle-loop, while the latter forms a stem-loop (*Chetverina et al., 2017*). In the studies reported

here, we have investigated how these two different topologies impact the local chromatin organization. We have also determined whether circle-loops and stem-loops alter the ability of boundary elements to define units of independent gene activity and insulate against regulatory interactions between neighboring TADs.

### *nhomie* deletion disrupts the *eve* TAD

As would be predicted from many different studies (*Cavalheiro et al., 2021*; *Chetverina et al., 2017*), deletion of the *nhomie* boundary and replacement with a control $\lambda$ *DNA* disrupts the *eve* TAD and alters the regulatory landscape. The MicroC profile shows that disruption of the *eve* TAD and the neighboring TADs is one-sided. Within the *eve* TAD, the subdomain linking *homie* and the nearby PRE to the *eve* promoter is unaffected. Likewise, the large TAD, TM, which encompassess both *TER94* and *pka-R2*, and is defined at one end by *homie* and at the other by an uncharacterized boundary element upstream of the *pka-R2* promoter, is intact (*Figure 2—figure supplement 1B*). In contrast, on the *nhomie* side of the *eve* TAD, the sub-TAD linking *nhomie* to the *eve* promoter is absent and is replaced by a less well-defined sub-TAD linking the *eve* promoter to an element near the *CG12134* promoter. However, the endpoint of the *eve* TAD is no longer distinct, and *eve* sequences are now crosslinked to the *eIF3j* sub-TAD and to sequences in the more distant TL sub-TADs TL3, TL2, and TL1. Consistent with these alterations in the physical organization of the *eve* and neighboring TADs, the *TER94* gene is still insulated from the *eve* enhancers. While sequences in the *eve* TAD physically interact with the gene closest to *nhomie*, *CG12134*, only the next gene over, *eIF3j*, is clearly activated by the *eve* enhancers. Since crosslinking between sequences in *CG12134* and the *eve* TAD is more frequent than crosslinking between the *eIF3j* sub-TAD TL3 and the *eve* TAD, it seems likely that *CG12134* is refractory to activation by the *eve* enhancers. This could be due to an incompatibility between the *CG12134* promoter and the *eve* enhancers. Alternatively, the promoter may not be active at this stage. While *eIF3j* is activated by the *eve* enhancers in stage 5 embryos in the *nhomie* deletion, the stripe enhancers located upstream of the *eve* gene drive a higher level of expression than those located downstream. Two factors in addition to the effects of distance could potentially account for this finding. Since the *eve* promoter is located between the downstream enhancers and the *eIF3j* gene, activation of *eIF3j* by the downstream enhancers could be suppressed by promoter competition. Alternatively, or in addition, the sub-TAD formed between the 3' PRE/*homie* and the *eve* promoter/proximal PRE (*Fujioka et al., 2008*) could tend to isolate the downstream *eve* stripe enhancers from interactions with *eIF3j*.

### Topology impacts local 3D genome organization and the potential for regulatory interactions

In boundary bypass experiments using endogenous fly boundaries, the ability of the upstream enhancers to activate the downstream reporter depended on the topology of the loop generated by the paired boundaries (*Kyrchanova et al., 2008a*). Activation is observed for stem-loops, as this configuration brings the upstream enhancers into close proximity with the downstream reporter. In contrast, the enhancers and downstream reporter are not brought into contact when the topology is a circle-loop. As would be predicted from these bypass experiments, the stem-loop formed by the head-to-tail pairing of *nhomie* and *homie* physically isolates the *eve* TAD from its neighbors, and this is reflected in the low density of contacts between sequences in *eve* and the neighboring TADs (*Figure 2A*). Conversely, the TADs that flank *eve* are brought together, and contacts between them generate the plume that is observed above the *eve* volcano triangle.

   The physical isolation afforded by the stem-loop topology is lost when the *eve* TAD is converted to a circle-loop either by inverting the *nhomie* boundary or by replacing *nhomie* with the *homie* boundary inserted in the forward direction (*Figure 6*). In the former case, head-to-tail *nhomie:homie* pairing generates a circle-loop. In the latter case, head-to-head pairing of *homie* (inserted in the forward orientation) with endogenous *homie* generates a circle-loop. The alteration in the local 3D organization induced by the conversion of *eve* to a circle-loop is evident from the changes in the MicroC contact pattern. Instead of being isolated from neighboring TADs, the *eve* TAD interacts not only with its immediate neighbors, but also with more distant TADs. As a result, the plume of enhanced contacts linking TM to TL, TK, and TJ is absent and is replaced by contacts between these TADs and the *eve* TAD.

As might be expected from the MicroC contact patterns, the conversion to a circle-loop topology is accompanied by alterations in regulatory interactions between *eve* and the genes in the neighboring TADs (*Figure 7*, *Figure 4—figure supplement 2*). Unlike the *nhomie forward* replacement, the *eve* stripe enhancers in both of the circle-loop replacements are able to weakly activate expression of two neighboring genes, *eIF3j* and *TER94*. This pattern of activation mirrors the enhancement in contacts between the *eve* TAD and the neighboring TL and TM TADs evident in MicroC experiments. Thus, though *eIF3j* and *TER94* are clearly shielded from the *eve* enhancers when the *eve* TAD is a circle-loop, a greater degree of isolation from the action of the *eve* enhancers is afforded when the *eve* TAD is a stem-loop.

In addition to reducing insulation from regulatory interactions with genes in neighboring TADs, the circle-loop topology impacts the functioning of the *eve* gene (*Figure 5B*). For both *nhomie reverse* and *homie forward*, the frequency of multiple denticle band defects compared to the *nhomie forward* control is enhanced in a sensitized genetic background. While we did not detect any obvious reductions in the *eve* stripes in blastoderm stage embryos, the circle-loop provides less insulation than the stem-loop, and it is possible that the neighboring promoters suppress *eve* expression by competing for the *eve* enhancers. Another (nonmutually exclusive) possibility comes from the studies of *Yokoshi et al., 2020*, who used live imaging to examine the effects of flanking a reporter with the *nhomie* and *homie* boundaries. In their experiments, reporter expression was enhanced over twofold when the reporter was flanked by *nhomie* and *homie*; however, the enhancement was greater when the paired boundaries formed a stem-loop than when they formed a circle-loop.

## Boundary:boundary pairing can generate stem-loops and circle-loops

Our manipulations of the *nhomie* boundary support the notion that TADs can have two different loop topologies, stem-loop and circle-loop, and these topologies impact their physical and biochemical properties. Since circle-loop TADs cannot be generated in the popular cohesin loop extrusion/CTCF roadblock model for the sculpting the 3D genome, it would be important to know whether there are other unambiguous examples of loops with either a stem-loop or circle-loop topology besides those described here. As discussed above, the available evidence suggests that the boundaries in the *Abd-B* regulatory domains pair with their neighbors head-to-head, and thus form circle-loops. While the contact pattern between neighbors in the *Abd-B* region fit with that expected for an array of circle-loop TADs, this has not been confirmed by examining the MicroC profiles before and after manipulating the boundary elements in this region. For this reason, we sought unambiguous examples of chromatin loops generated by the orientation-dependent pairing of endogenous TAD boundaries that have either a stem-loop or a circle-loop topology. The collection of meta-loops described recently by *Mohana et al., 2023* provide one such test, as many appear to be generated by the pairing of TAD boundaries, and their local interaction profiles are easily interpreted.

Shown in *Figure 8A* is a 2.8 Mb meta-loop on chromosome 2L generated by the pairing of two TAD boundaries, labeled blue and purple (block arrows). The pairing of the blue and purple TAD boundaries brings sequences in the TADs flanking the two boundaries into contact, and this generates two rectangular boxes of interaction indicated by the arrows (see also blue double arrows in the diagram on the right). In the rectangular box on the upper left of the contact map, sequences in the TAD containing *CG33543*, *Obp22a,* and *Npc2a* located just upstream of the blue boundary are crosslinked to sequences upstream of the purple boundary in the TAD containing the *fipi* gene. In the rectangular box on the lower right, sequences in a small TAD downstream of the blue boundary (which contains *Nplp4* and *CG15353*) are crosslinked to sequences in a TAD downstream of the purple boundary (which contains *CG3294* and *slf*). As shown in the diagram, this pattern of interaction (upstream-to-upstream and downstream-to-downstream) indicates that the blue and purple TAD boundaries pair with each other head-to-head, and this orientation generates a large loop with a circle-loop topology.

The 2.2 Mb meta-loop on chromosome 3L in *Figure 8B* is more complicated in that it is generated by four TAD boundaries (indicated by blue, brown, green, and purple arrows). The blue and brown boundaries separate a small TAD containing *CG7509* from two larger TADs, while the green and blue boundaries define the endpoints of a TAD containing the most distal promoter (blue arrowhead) of the *Mp* (*Multiplexin*) gene. As indicated in the accompanying diagram, the brown and green boundaries pair with each other head-to-tail as do the blue and purple boundaries. Pairing of the brown and green boundaries generates a large ~2.2 Mb stem-loop that brings sequences in the TAD

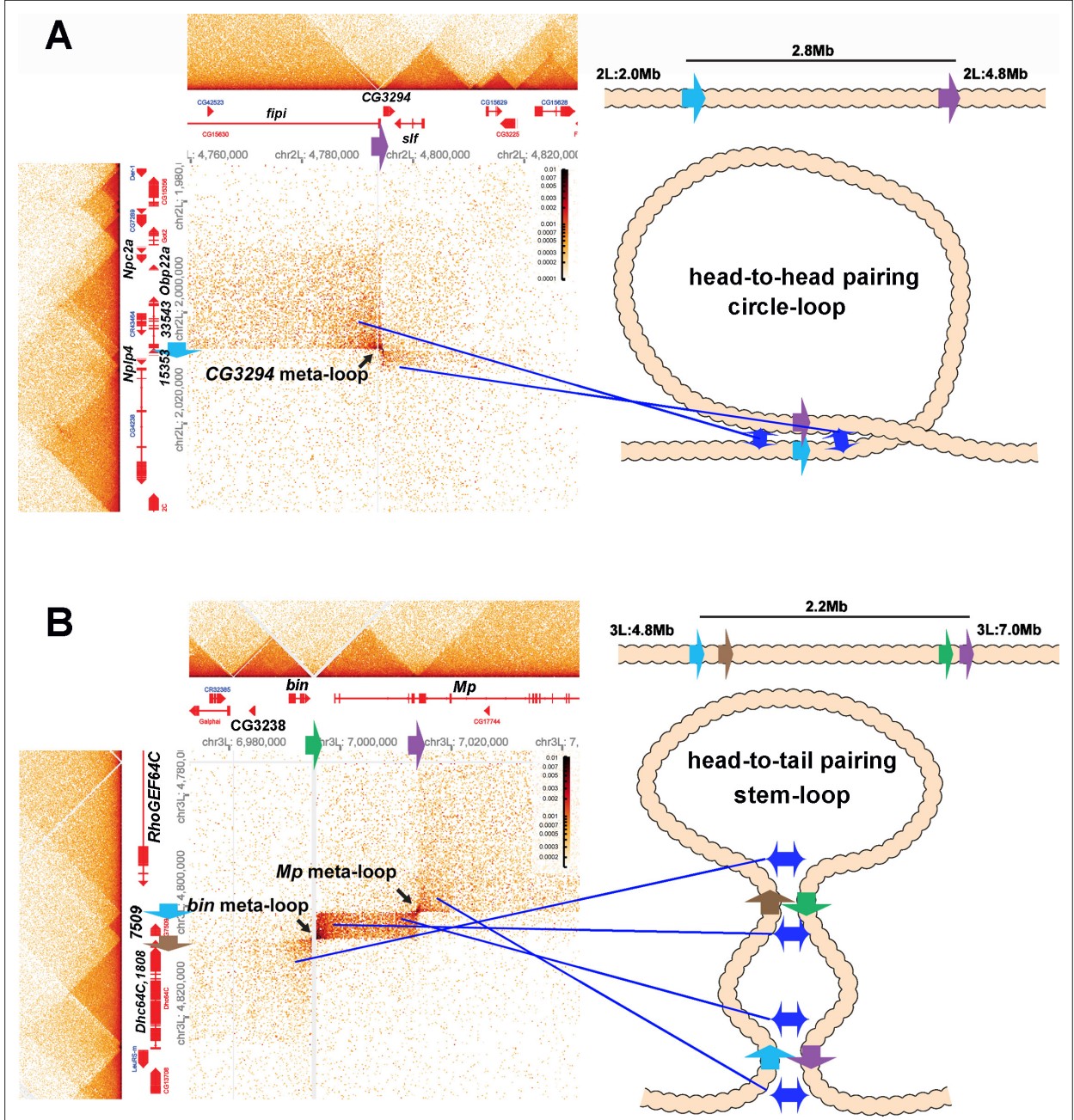

**Figure 8.** Circle-loop and stem-loop meta-loops. (**A**) *CG3294* circle-loop meta-loop. In this meta-loop, a TAD boundary (blue arrow) located at ~2.0 Mb on chromosome 2L pairs head-to-head with a TAD boundary (purple arrow) located ~2.8 Mb away. As indicated in the diagram, head-to-head pairing generates a circle-loop. In the circle-loop topology, the TAD upstream of the blue boundary is brought into contact with the TAD upstream of the purple boundary, as indicated the diagram (blue double arrows). This generates a rectangular box of enhanced contacts between sequences in the TAD containing the *CG33543*, *Obp22a*, and *Npc2a* genes and sequences in a TAD that contains the *fipi* gene. This box is located on the upper left of the contact map (above and to the left of the black arrow). Sequences in TADs downstream of the blue and purple boundaries are also linked, and this generates a small rectangular box representing sequences in the small *Nplp4* and *CG15353* TAD ligated to sequences in the TAD containing *CG3294* and *slf* (below and to the right of the black arrow). (**B**) The *bin/Mp* meta-loops on the left arm of the third chromosome are generated by the head-to-tail pairing of two sets of boundaries, indicated by the blue, brown, green, and purple arrows. Pairing of the brown and green boundaries generates an ~2.2 Mb stem-loop. Sequences in the TAD downstream of the brown boundary (which contains the *Dhc64C* and *CG1808* genes) are linked to sequences in the TAD upstream of the green boundary (which contains *CG2328* and *bin*). This generates the rectangular box of enhanced contacts on the lower left of the contact map. Pairing of the blue and purple boundaries head-to-tail generates a small stem-loop 'bubble' (see diagram). This bubble brings sequences in the TAD containing the most distal *Mp* promoter (blue arrowhead) into contact with sequences in the small TAD containing *CG7509* (see diagram on the right). Interactions between these two TADs generates the small rectangular box of enhanced contacts in the center of the contact

*Figure 8 continued*

map. The head-to-tail pairing of the blue and purple boundaries also bring sequences in the TAD upstream of the blue boundary that contains the *RhoGEF64C* gene into contact with the TAD containing one of the internal *Mp* promoters (black arrowhead). This interaction generates the box of enhanced contacts in the upper-right portion of the contact map. The bin size for each panel is 200 bp; embryos are 12–16 hr old.

The online version of this article includes the following figure supplement(s) for figure 8:

**Figure supplement 1.** MicroC patterns of DNA segments on the left and right arm of chromosome 2.

downstream of the brown boundary, which contains the *Dhc64C* and *CG1808* genes, into contact with sequences in the TAD upstream of the green boundary, which contains the *CG3238* and *bin* (*binou*) genes. This linkage generates the rectangular box of enhanced contacts in the lower-left corner of the contact map. Pairing of the blue and purple boundaries generates a small stem-loop bubble that links sequences in the *CG7509* TAD to the TAD containing the *Mp* distal promoter (blue arrowhead). This connection generates a small rectangular box of enhanced crosslinking in the center of the contact map. In addition, sequences upstream of the blue boundary and downstream of the purple boundary are brought into contact by the head-to-tail pairing of these two boundaries. This generates a third rectangular box of enhanced physical contact in the upper-right corner of the contact map that links sequences in the TAD containing the *RhoGEF64C* gene to sequences in the TAD containing the internal *Mp* promoter (black arrowhead). Note that the positioning of the lower-left and upper-right rectangular boxes of enhanced contact in the *bin-MP* meta-loop is the mirror image of the rectangular boxes of enhanced contact for the *CG3294* meta-loop.

## Stem-loops versus circle-loops

The results we have reported here demonstrate that boundary:boundary pairing in flies generates loops that can have either a stem-loop or a circle-loop topology. An important question is, what is the relative frequency of stem-loops versus circle-loops in the fly genome? The MicroC contact patterns for the stem-loop and circle-loop versions of the *eve* TAD are quite distinct. The former is a volcano triangle with a plume while the latter is a volcano triangle flanked by clouds. A survey of the MicroC contact patterns elsewhere in the non-repetitive regions of the fly genome indicates that volcanoes with plumes are rare (~30). For example, there are two volcano triangles with plumes in the *Antennapedia* complex, and they encompass the *deformed* and *fushi-tarazu* genes (***Levo et al., 2022***). However, since most of the 'euchromatic' regions of the fly genome are assembled into TADs whose MicroC profiles resemble that observed for *eve* circle-loops and the *Abd-B* region of BX-C, it is possible that much of the fly genome is assembled into circle-loops, not stem-loops.

While this suggestion is consistent with the available data, it is based on contact patterns between neighboring TADs, and important caveats remain. For one, the contact patterns between neighboring TADs can deviate in one way or another from that seen in the *Abd-B* region. For example, there are TADs in which interactions with one set of neighbors appear to be suppressed as expected for stem-loops, but the classic plume is absent, as interactions are not suppressed with the other neighbors (c.f., ***Figure 8—figure supplement 1A***). In other cases, there is a series of complicated TAD-TAD interactions topped by a rectangular plume (***Figure 8—figure supplement 1***, purple arrow). For this reason, it will not be possible to draw firm conclusions about the frequency of stem-loops versus circle-loops genome-wide until the relative orientation of the paired boundaries themselves can be determined directly. On the other hand, it is clear from our studies that both classes of TADs must exist in flies. If, as seems likely, a significant fraction of the TADs genome-wide are circle-loops, this would effectively exclude cohesin-based loop extrusion as a general mechanism for TAD formation in flies. In addition, though stem-loops could be generated by a cohesin-dependent mechanism, it is unlikely that this mechanism is operational in flies, as we have shown here and in ***Bing et al., 2024*** that stem-loops in flies are formed by orientation-dependent boundary:boundary pairing.

Another important question is whether our findings have any relevance to the formation and topology of TADs in mammals. In the most common version of the loop extrusion model, the mammalian genome is assembled into an alternating pattern of stem-loops and unanchored loops (***Davidson and Peters, 2021***; ***Higashi and Uhlmann, 2022***; ***Perea-Resa et al., 2021***). In this case, one might expect to observe volcano triangles topped by plumes alternating with DNA segments that have a considerably lower density of internal contacts. However, this crosslinking pattern is not observed in published MicroC data sets (***Hsieh et al., 2020***; ***Krietenstein et al., 2020***). Instead of an alternating

pattern of high-density TAD triangles separated by regions of low-density contacts, the TAD triangles are generally linked to both neighbors, just as in flies. Moreover, also like in *Drosophila*, there are few stem-loop volcano TADs topped by plumes. Instead, the crosslinking pattern between neighboring TADs appears similar to that observed for circle-loops in flies. Of course, one problem with these MicroC studies is that the resolution may not be sufficient to detect volcanoes with plumes or the other features predicted by the loop-extrusion model. However, there are no obvious volcanoes with plumes in the much higher resolution RCMC studies of *Goel et al., 2023*. Instead, the MicroC profiles most closely resemble those seen in the *Abd-B* region of BX-C (c.f. the *Ppm1g* locus in Figure 4 of *Goel et al., 2023*). Moreover, compromising cohesin activity has minimal impact on the TADs in this region of the mouse genome, as evidenced from the MicroC pattern before and after knockdown. Based on these observations, one can reasonably question whether cohesin-mediated loop extrusion is deployed in mammals as the mechanism for not only generating TADs but also determining TAD boundaries. Clearly, validation of the loop-extrusion/CTCF road-block model as currently formulated will require a direct demonstration that mammalian TADs are exclusively either stem-loops or unanchored loops, and that the endpoints are always (or almost always) determined by CTCF roadblocks.

## TADs and A/B compartmentalization

A/B compartmentalization has been proposed as a mechanism for subdividing the chromosome into discrete domains that is independent of cohesin-mediated loop extrusion and CTCF. In this model, shared biochemical/biophysical properties that reflect the relative transcriptional state of each chromosomal segment drive block polymer co-segregation into a series of discrete domains (*Rowley and Corces, 2018*; *Rowley et al., 2017*). While previous studies suggested that the A and B compartments represented Mb-scale DNA segments, in more recent studies, *Harris et al., 2023* found that the average compartment size is on the order of 12 kb. Not only is this much smaller than originally suggested, it is also similar in size to that of most TADs in the *Drosophila* genome, including the *eve* TAD. Moreover, in their studies (and also in our data sets), there is close to a one-to-one correspondence between the linear arrangement of individual TADs along the chromosome and the DNA segments that are thought to assemble into discrete domains by co-polymer segregation. This close connection to TADs is also reflected in the patchwork patterns of interacting chromatin domains that are visualized in studies on A/B compartments.

According to this newer version of the compartment model, the chromatin state of each DNA segment determines not only whether it will assemble into a discrete domain, but also how the resulting domain interacts with next-door neighbors, next-next-door neighbors, etc. However, this model does not appear to fit with several of our findings. To begin with, the sequences included in the *eve* TAD and its patterns of interaction with neighboring TADs are essentially the same in NC14 embryos as they are in 12–16 hr embryos. In the former case, *eve* is transcriptionally poised (*Chen et al., 2013*), probably in most or all nuclei, while in the latter case, the entire *eve* TAD is silenced by a PcG-dependent mechanism in all but a few nuclei (*Nègre et al., 2010*). However, this transition from potentially active to silenced does not impact the *eve* TAD, nor does it alter how the *eve* TAD interacts with the neighboring TADs that are (mostly) transcriptionally active at both stages of development. Similarly, in the meta-loops we have examined, the transcriptional state of the TADs and their contact patterns with their neighbors are not consistent with a strict partitioning of chromosomal segments into one of two compartments. For example, in the *CG3294* meta-loop (*Figure 8A*), the three genes (*CG33534*, the odorant binding gene *Obp22a* and the *Npc2a* gene) that comprise the TAD upstream of the blue boundary would be predicted to be in the same chromatin state; however, while *CG33534* and *Obp22a* are not expressed in embryos, the *Npc2a* gene is expressed at high levels during embryogenesis and should partition into a separate TAD. The TAD downstream of the blue boundary contains two transcriptionally repressed genes (*CG15353* and *Nplp4*) in 12–16 hr embryos. This TAD physically interacts with the TAD downstream of the purple boundary that contains *CG3294* and *slf*. According to the A/B compartment model, the *CG15353* and *Nplp4* TAD interacts with the *CG3294* and *slf* TAD because the chromatin in these two TADs share biochemical/biophysical properties that are characteristic of the inactive B compartment. However, unlike *CG15353* and *Nplp4*, both *slf* and *CG3294* are expressed in 12–16 hr embryos, *slf* at a high level and *CG3294* at a low level.

Likewise, if block-polymer co-segregation is the determining factor for both TAD formation and the patterns of TAD:TAD interactions, then our manipulations of the *nhomie* boundary should have only a

minimal impact on the MicroC contact maps, unless there are significant changes in the transcriptional status of *eve* and its neighbors. In the *lambda* DNA replacement, the left endpoint of the WT *eve* TAD (the normal location of *nhomie*) in 12–16 hr embryos is lost, and instead it appears to map primarily to the right or left boundaries of the TL-3 sub-TAD. TL-3 contains the *eIF3j* gene, which is expressed at high levels throughout the embryo at this stage, while *eve* itself is silenced by a PcG-dependent mechanism in all but a small number of cells. The *nhomie reverse* and *homie forward* replacements restore the *eve* TAD. This means that in this instance, TAD formation is mediated by the pairing of the two replacement boundaries with *homie* (as demonstrated by the viewpoints in *Figure 6*) and not by partitioning into an A or B compartment. The replacements also alter interactions between *eve* and the neighboring TADs. Unlike in WT where the *eve* TAD is physically isolated from its neighbors, the *eve* TAD interacts with both neighbors in these two replacements. However, the genes in the neighboring TADs do not share the same biochemical/biophysical state—they are active in 12–16 hr embryos, and so should segregate into the A compartment, while *eve* is inactive and should segregate into the B compartment.

While these observations are inconsistent with a model in which block-polymer co-segregation is responsible for the formation of TADs and determining the pattern of TAD:TAD interactions, this does not rule out a different role, namely in augmenting the insulating activity of boundary elements. One of the defining properties of boundary elements is to restrict the activities of enhancers and silencers, and this helps ensure that the chromatin within a given TAD shares the same biochemical and biophysical properties. The shared biochemical/biophysical properties could in turn enhance the segregation of the chromatin into different compartments, and thus mediate some of the changes we observe. As we cannot rule out the possibility that such biophysical forces augment the functional properties of boundaries, we should add them to the list of downstream events that are dependent upon boundary–boundary interactions and the specific topologies that they can induce. So, while compartment co-segregation may well play a role, on multiple length scales, in mediating the effects of boundaries and other regulatory elements on gene expression and chromosome topology, it certainly cannot 'replace' their functional properties as an explanation for those effects.

## Materials and methods
### Creation of *nhomie* deletion flies

To modify *nhomie* at the *eve* locus, we used recombinase-mediated cassette exchange (*Bateman et al., 2006*). First, we inserted two closely positioned attP sites using CRISPR. The donor plasmid for this was constructed as follows. First, a mini-*white* (*mw*) gene with Glass binding sites (*Fujioka et al., 1999*) was inserted into pBlueScript. From the standard *mw* gene, the Wari insulator (*Chetverina et al., 2008*) was deleted. Then, two 102 bp attP sequences (*Venken et al., 2011*) were inserted, one just 5′ of the Glass binding sites and the other at the 3′ end of the modified *mw*, creating the plasmid pP-attPx2-*mw*. 5′- and 3′-homologous arms were added to both ends. Two gRNA sequences were cloned into plasmid pCFD4 (*Port et al., 2014*) (Addgene). The donor and gRNA plasmids were injected into a Cas9 line (y[1] M{vas-Cas9.S}ZH-2A w[1118], Bloomington Drosophila Stock Center). This chromosomal modification resulted in one attP site being inserted in the intron of *CG12134*, and the other being inserted between the *eve* 7-stripe enhancer and the 3+7 stripe enhancer. This also deleted 2.2 kb of endogenous sequence, including *nhomie* and the *eve* 7-stripe enhancer.

After identifying a successful insertion (NattPmw), *mw* was replaced by each of the following using RMCE: (1) the previously deleted 2.2 kb, restoring *nhomie* and the *eve* 7-stripe enhancer, to create 'wild-type *nhomie*" (*nhomie forward*), (2) the same 2.2 kb sequence, but with 600 bp of phage $\lambda$ DNA in place of 600 bp *nhomie* ($\lambda$ DNA), (3) the same 2.2 kb sequence, but with 600 bp *nhomie* inverted (*nhomie reverse*), and (4) the same 2.2 kb sequence, but with 600 bp *nhomie* replaced by a copy of ~600 bp *homie* in its native orientation in the chromosome (*homie forward*). Each of these changes was confirmed by sequencing of genomic DNA from the transgenic fly lines.

### Analysis of embryonic cuticle patterns and *in situ* hybridization

To identify defects in developing embryos, embryos were collected for 2.5 hr, and allowed to develop for an additional 20–21 hr at 25°C. Embryos were dechorionated and mounted in a 1:1 mixture of Hoyer's medium and lactic acid. Mounted embryos were left at room temperature (RT) until they

cleared (12–14 days), and the patterns of ventral abdominal denticles were examined and tallied as follows. Loss of at least one-fifth of a denticle band (in A1-A8) was counted as 'missing'. Fused denticle bands, which rarely occurred, were also counted as a 'missing' band. Minor defects such as those within individual denticle rows were not counted.

Digoxigenin (DIG) *in situ* hybridization was performed using DIG-labeled anti-sense RNA against *CG12134*, *eIF3j*, and *TER94*. RNA expression was visualized using alkaline phosphatase-conjugated anti-DIG antibody (Roche), using CBIP and NBT as substrates (Roche). Each set of experiments was carried out in parallel to minimize experimental variation. Representative expression patterns are shown in each figure.

## HCR-FISH

The sequences of target genes were obtained from FlyBase (https://www.flybase.org/; *Gramates et al., 2022*). To design probes, the target gene sequences were submitted to the Molecular Instruments probe design platform (https://molecularinstruments.com/hcr-rnafish; *Choi et al., 2016*), with parameters set to a 35 probe set size for *Drosophila melanogaster*. A similar method was designed based on published smFISH methods (*Little and Gregor, 2018*; *Trcek et al., 2017*). 100–200 flies were placed in a cage with an apple juice plate at the bottom. For early stages, the embryos were collected for 7 hr, while for later-stage embryos, collections were overnight. Embryos from each plate were washed into collection mesh and dechorionated in bleach for 2 min, then fixed in 5 mL of 4% paraformaldehyde in 1× PBS and 5 mL of heptane for 15 min with horizontal shaking. The paraformaldehyde was then removed and replaced with 5 mL methanol. The embryos were then devitellinized by vortexing for 30 s and washed in 1 mL of methanol twice. Methanol was then removed and replaced by PTw (1× PBS with 0.1% Tween-20) through serial dilution as 7:3, 1:1, and 3:7 methanol:PTw. The embryos were washed twice in 1 mL of PTw and pre-hybridized in 200 µL of probe hybridization buffer for 30 min at 37°C. 0.4 pmol of each probe set were added to the embryos in the probe hybridization buffer, and the embryos were incubated at 37°C for 12–14 hr. The embryos were then washed 3× with probe wash buffer at 37°C for 30 min and 2× with 5× SSCT (5× SSC + 0.1% tween) at RT for 5 min. Then the embryos were pre-amplified with 300 µL amplification buffer for 10 min at 25°C. Meanwhile, 6 pmol of hairpin h1 and h2 were snap-cooled separately (95°C for 90 s, cool to RT with 0.1°C drop per second), and then mixed in 100 µL of amplification buffer at RT. After that, the pre-amplification solution was removed from the embryos, and 100 µL of hairpin h1/h2 mix were added to the embryos. Next, the embryos were incubated for 12–14 hr at RT in the dark. To remove excess hairpins, the embryos were washed in SSCT as follows: 2× for 5 min, 2× for 30 min, and 5× for 5 min. Then, the embryos were washed with 1 mL PTw for 2 min and stained with DAPI/Hoechst at 1 µg/mL for 15 min at RT in the dark. The embryos were then washed with PTw 3× for 5 min. Finally, the embryos were mounted on microscope slides with Vectashield and a #1.5 coverslip for imaging.

## Imaging, image analysis, and statistics

Embryos from HCR-FISH were imaged using a Nikon A1 confocal microscope system, with a Plan Apo ×20/0.75 DIC objective. Z-stack images were taken at interval of 2 µm, 4× average, 1024 × 1024 resolution, and the appropriate laser power and gain were set for 405, 488, 561, and 640 channels to avoid overexposure. Images were processed using ImageJ, and the maximum projection was applied to each of the stack images. To determine the presence of stripes in early embryos, multi-channel images were first split into single channels and the stripe signal was highlighted and detected by the MaxEntropy thresholding method. GraphPad Prism was used for data visualization and statistical analysis. Two-way ANOVA with Tukey's multiple comparisons test for each pair of groups was used to determine the statistical significance for the percentage of embryos carrying stripes in *eIF3j* and *TER94* channels in each group.

## MicroC library construction for the *nhomie* replacements

Embryos were collected on yeasted apple juice plates in population cages for 4 hr, incubated for 12 hr at 25°C, then subjected to fixation as follows. Embryos were dechorionated for 2 min in 3% sodium hypochlorite, rinsed with deionized water, and transferred to glass vials containing 5 mL PBST (0.1% Triton-X100 in PBS), 7.5 mL n-heptane, and 1.5 mL fresh 16% formaldehyde. Crosslinking was carried out at RT for exactly 15 min on an orbital shaker at 250 rpm, followed by addition of 3.7 mL 2 M

Tris-HCl pH 7.5 and shaking for 5 min to quench the reaction. Embryos were washed twice with 15 mL PBST and subjected to secondary crosslinking. Secondary crosslinking was done in 10 mL of freshly prepared 3 mM final DSG and ESG in PBST for 45 min at RT with passive mixing. The reaction was quenched by addition of 3.7 mL of 2 M Tris-HCl pH7.5 for 5 min, washed twice with PBST, snap-frozen, and stored at –80°C until library construction.

Micro-C libraries were prepared as previously described (*Batut et al., 2022*) with the following modifications: 50 μL of 12–16 hr embryos were used for each biological replicate. 60U of MNase was used for each reaction to digest chromatin to a mononucleosome:dinucleosome ratio of 4. Libraries were barcoded, pooled, and subjected to paired-end sequencing on an Illumina Novaseq S1 100nt Flowcell (read length 50 bases per mate, 6-base index read).

Two or more independent biological replicates were sequenced for each genotype. For each replicate, >1000 embryos were used, and ~250M reads sequenced. Post-sequencing QC analysis was done for every sample, and the QC reports are available along with the sequence data in GEO (GSE263270). The raw sequencing data are also available in GSE263270 for use in further bioinformatics analysis. The figures present the merged data from all independent biological replicates for each genotype. The total read numbers in the merged data are very similar for each genotype (~500M reads). For NC14 embryo MicroC (*Figure 2*), public data sets GSE171396 and GSE173518 were used (*Batut et al., 2022*; *Levo et al., 2022*).

## MicroC data processing

MicroC data for *D. melanogaster* were aligned to custom genomes edited from the Berkeley Drosophila Genome Project (BDGP) Release 6 reference assembly (*dos Santos et al., 2015*) with BWA-MEM (*Li and Durbin, 2009*) using parameters **-S -P -5 -M**. The resultant BAM files were parsed, sorted, de-duplicated, filtered, and split with Pairtools (https://github.com/open2c/pairtools; *Golob-orodko, 2024*). We removed pairs where only half of the pair could be mapped, or where the MAPQ score was less than three. The resultant files were indexed with Pairix (https://github.com/4dn-dcic/pairix; *Lee, 2024*). The files from replicates were merged with Pairtools before generating 100 bp contact matrices using Cooler (*Abdennur and Mirny, 2020*). Finally, balancing and Mcool file generation were performed with Cooler's Zoomify tool.

Virtual 4C profiles were extracted from individual replicates using FAN-C (*Kruse et al., 2020*) at 400 bp resolution. The values were summed across replicates and smoothed across three bins (1.2 kb). The *homie* viewpoint was set to the 549nt *homie* sequence that was defined in previous studies (*Fujioka et al., 2016*; *Fujioka et al., 2009*).

## Acknowledgements

We thank Gordon Grey for running the fly food facility at Princeton, members of the Lewis Sigler Genomics Core facility for their invaluable assistance with DNA sequencing, and Qing Liu for excellent technical assistance. We would also like to thank members of MOL431 for creative input. Special thanks to Olga Kyrchanova, Daria Chetverina, Maksim Erokhin, Pavel Georigev, Tsutomu Aoki, Girish Deshpande, Airat Ibragimov, Sergey Ryabichko, Yuri Pritykin, Alex Ostrin, Xinyang Bing, Xiao Li, and Mike Levine for stimulating discussions and sharing unpublished data.

## Additional information

### Funding

| Funder | Grant reference number | Author |
|---|---|---|
| New Jersey Commission on Cancer Research | COCR23PDF011 | Wenfan Ke |
| Histochemical Society | Keystone Grant | Wenfan Ke |
| National Institute of General Medical Sciences | R35 GM126975 | Paul Schedl |

| Funder | Grant reference number | Author |
|---|---|---|
| National Institute of General Medical Sciences | R01 GM137062 | James B Jaynes |

The funders had no role in study design, data collection and interpretation, or the decision to submit the work for publication.

### Author contributions
Wenfan Ke, Conceptualization, Formal analysis, Funding acquisition, Investigation, Methodology; Miki Fujioka, Conceptualization, Formal analysis, Investigation, Methodology; Paul Schedl, Conceptualization, Supervision, Funding acquisition, Writing - original draft, Writing - review and editing; James B Jaynes, Conceptualization, Supervision, Funding acquisition, Writing - review and editing

### Author ORCIDs
Wenfan Ke https://orcid.org/0000-0002-7047-5445
Paul Schedl https://orcid.org/0000-0001-5704-2349
James B Jaynes http://orcid.org/0000-0001-7943-794X

Reviewer #1 (Public Review): https://doi.org/10.7554/eLife.94114.3.sa1
Reviewer #2 (Public Review): https://doi.org/10.7554/eLife.94114.3.sa2
Author response https://doi.org/10.7554/eLife.94114.3.sa3

## Additional files

### Supplementary files
• MDAR checklist

### Data availability
Sequence data are available at GEO GSE263270. Confocal images are available on Open Science Framework at https://doi.org/10.17605/OSF.IO/6PYBM.

The following datasets were generated:

| Author(s) | Year | Dataset title | Dataset URL | Database and Identifier |
|---|---|---|---|---|
| Ke W, Fujioka M, Schedl P, Jaynes J | 2024 | Chromosome Structure II: Stem-loops and circle-loops | http://www.ncbi.nlm.nih.gov/geo/query/acc.cgi?acc=GSE263270 | NCBI Gene Expression Omnibus, GSE263270 |
| Ke W | 2024 | Stem-loop and circle-loop TADs generated by directional pairing of boundary elements have distinct physical and regulatory properties | https://osf.io/6pybm/ | Open Science Framework, 6pybm/ |

The following previously published datasets were used:

| Author(s) | Year | Dataset title | Dataset URL | Database and Identifier |
|---|---|---|---|---|
| Bing X, Batut P, Levine M | 2022 | Genome organization controls transcriptional dynamics during development | http://www.ncbi.nlm.nih.gov/geo/query/acc.cgi?acc=GSE171396 | NCBI Gene Expression Omnibus, GSE171396 |
| Bing X, Levo M, Raimundo J, Levine M | 2022 | Transcriptional coupling of distant regulatory genes in living embryos | http://www.ncbi.nlm.nih.gov/geo/query/acc.cgi?acc=GSE173518 | NCBI Gene Expression Omnibus, GSE173518 |

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

# Appendix 1

## Appendix 1—key resources table

| Reagent type (species) or resource | Designation | Source or reference | Identifiers | Additional information |
|---|---|---|---|---|
| Gene (*Drosophila melanogaster*) | *eve* | FlyBase | FBgn0000606 | |
| Gene (*D. melanogaster*) | CG12134 | FlyBase | FBgn0033471 | |
| Gene (*D. melanogaster*) | eIF3j | FlyBase | FBgn0027619 | |
| Gene (*D. melanogaster*) | TER94 | FlyBase | FBgn0286784 | |
| Genetic reagent (*D. melanogaster*) | y1 M{vas-Cas9}ZH-2A w1118/FM7c | Bloomington Drosophila Stock Center | 51323 | |
| Recombinant DNA reagent (plasmid) | pCFD4-U6:1_U6:3tandemgRNAs | Addgene | 49411 | |
| Chemical compound, drug | n-Heptane | Fisher Chemical | O3008-4 | |
| Chemical compound, drug | Paraformaldehyde 20% solution, EM Grade | Electron Microscopy Sciences | 15713S | |
| Chemical compound, drug | Formaldehyde, 16%, methanol free, Ultra Pure | Polysciences Inc | 18814-10 | |
| Chemical compound, drug | PBS – phosphate-buffered saline (10×) pH 7.4, RNase-free | Thermo Fisher | AM9624 | |
| Chemical compound, drug | Tween 20 | Sigma | P1379 | |
| Chemical compound, drug | Triton X-100 | Bio-Rad | 161-0407 | |
| Chemical compound, drug | Tris base | Sigma | 11814273001 | |
| Chemical compound, drug | Methanol | Fisher Chemical | 203403 | |
| Chemical compound, drug | SSC, 20× | Thermo Fisher | 15557044 | |
| Chemical compound, drug | Formamide | Thermo Fisher | 17899 | |
| Chemical compound, drug | Dextran sulfate | Sigma | D8906 | |
| Chemical compound, drug | Salmon Sperm DNA | Thermo Fisher | AM9680 | |
| Chemical compound, drug | Ribonucleoside Vanadyl Complex | NEB | S1402S | |
| Chemical compound, drug | Nuclease-free BSA | Sigma | 126609 | |
| Chemical compound, drug | Triethylammonium acetate | Sigma | 625718 | |
| Chemical compound, drug | dGTP (100 MM) | VWR | 76510-208 | |
| Chemical compound, drug | dTTP (100 MM) | VWR | 76510-224 | |
| Chemical compound, drug | Lonza NuSieve 3:1 Agarose | Thermo Fisher | BMA50090 | |
| Other | T4 DNA ligase | NEB | M0202L | Enzyme |

*Appendix 1 Continued on next page*

*Appendix 1 Continued*

| Reagent type (species) or resource | Designation | Source or reference | Identifiers | Additional information |
|---|---|---|---|---|
| Chemical compound, drug | Biotin-11-dCTP | Jena Bioscience | NU-809-BIOX | |
| Chemical compound, drug | Biotin-14-dATP | Jena Bioscience | NU-835-BIO14 | |
| Commercial assay or kit | Qubit dsDNA HS Assay Kit | Life Technologies Corp. | Q32851 | |
| Chemical compound, drug | Atto 633 NHS ester | Sigma | 01464 | |
| Chemical compound, drug | Phase Lock Gel, QuantaBio - 2302830, Phase Lock Gel Heavy | VMR | 10847-802 | |
| Commercial assay or kit | NEBNext Ultra II DNA Library Prep Kit for Illumina | NEB | E7645S | |
| Commercial assay or kit | Ampure Xp 5 ml Kit | Thermo Fisher | NC9959336 | |
| Commercial assay or kit | Hifi Hotstart Ready Mix | Thermo Fisher | 501965217 | |
| Commercial assay or kit | Dynabeads MyOne Streptavidin C1 | Life Technologies Corp. | 65001 | |
| Chemical compound, drug | cOmplete, EDTA-free Protease Inhibitor Cocktail | Sigma | 11873580001 | |
| Chemical compound, drug | *N,N*-Dimethylformamide | Sigma | 227056 | |
| Chemical compound, drug | Potassium acetate solution | Sigma | 95843 | |
| Chemical compound, drug | DSG (disuccinimidyl glutarate) | Thermo Fisher | PI20593 | |
| Other | T4 Polynucleotide Kinase – 500 units | NEB | M0201S | Enzyme |
| Other | DNA Polymerase I, Large (Klenow) Fragment – 1000 units | NEB | M0210L | Enzyme |
| Commercial assay or kit | End-it DNA End Repair Kit | Thermo Fisher | NC0105678 | |
| Other | Proteinase K recomb. 100 mg | Sigma | 3115879001 | Enzyme |
| Other | Nuclease Micrococcal (s7) | Thermo Fisher | NC9391488 | Enzyme |
| Chemical compound, drug | EGS (ethylene glycol *bis*(succinimidyl succinate)) | Thermo Fisher | PI21565 | |
| Commercial assay or kit | Atto 565 NHS ester | Sigma | 72464 | |
| Commercial assay or kit | HCR RNA-FISH Custom Probe Set: eve | Molecular Instruments | Custom probes | |
| Commercial assay or kit | HCR RNA-FISH Custom Probe Set: ter94 | Molecular Instruments | Custom probes | |
| Commercial assay or kit | HCR RNA-FISH Custom Probe Set: CG12134 | Molecular Instruments | Custom probes | |
| Commercial assay or kit | HCR RNA-FISH Custom Probe Set: eIF3j | Molecular Instruments | Custom probes | |
| Commercial assay or kit | HCR Amplifier B1, 488 | Molecular Instruments | Custom probes | |
| Commercial assay or kit | HCR Amplifier B2, 564 | Molecular Instruments | Custom probes | |
| Commercial assay or kit | HCR Amplifier B3, 647 | Molecular Instruments | Custom probes | |

*Appendix 1 Continued*

| Reagent type (species) or resource | Designation | Source or reference | Identifiers | Additional information |
|---|---|---|---|---|
| Commercial assay or kit | HCR Buffers | Molecular Instruments | Custom probes | |
| Commercial assay or kit | NEBNext Multiplex Oligos for Illumina | NEB | E7335S | |
| Software, algorithm | Fiji (ImageJ) | *Schindelin et al., 2012* | fiji.sc | |
| Software, algorithm | NIS element | Nikon | microscope.healthcare.nikon.com/products/software/nis-elements | |
| Software, algorithm | GraphPad Prism 8 | GraphPad Software | https://www.graphpad.com/ | |
| Software, algorithm | HiGlass | *Kerpedjiev et al., 2018* | https://higlass.io/app | |
| Software, algorithm | bwa | *Li and Durbin, 2009* | https://bio-bwa.sourceforge.net/ | |
| Software, algorithm | samtools | GitHub/open source | https://samtools.github.io | |
| Software, algorithm | pairsamtools | *Goloborodko et al., 2024* | https://github.com/mirnylab/pairsamtools | |
| Software, algorithm | pairix | *Lee, 2024* | https://github.com/4dn-dcic/pairix | |
| Software, algorithm | cooler | *Abdennur and Mirny, 2020; Abdennur, 2016* | https://github.com/open2c/cooler | |
| Software, algorithm | Miniconda | Anaconda | https://docs.conda.io/en/latest/miniconda/ | |
| Software, algorithm | Snakemake | GitHub/open source | https://snakemake.github.io | |

