## [Editor Report · eLife assessment]

This **valuable** work investigates the role of boundary elements in the formation of 3D genome architecture. The authors established a specific model system that allowed them to manipulate boundary elements and examine the resulting genome topology. The work yielded the first demonstration of the existence of stem and circle loops in a genome and confirms a model which had been posited based on extensive prior genetic work, providing insights into how 3D genome topologies affect enhancer–promoter communication. The evidence is **solid**, although the degree of generalization remains uncertain.

---

## [Referee Report · Reviewer #1 (Public Review)]

In this study, the authors engineer the endogenous left boundary of the *Drosophila* eve TAD, replacing the endogenous Nhomie boundary by either a neutral DNA, a wildtype Nhomie boundary, an inverted Nhomie boundary, or a second copy of the Homie boundary. They perform Micro-C on young embryos and conclude that endogenous Nhomie and Homie boundaries flanking eve pair with head-to-tail directionality to form a chromosomal stem loop. Abrogating the Nhomie boundary leads to ectopic activation of genes in the former neighboring TAD by eve embryonic stripe enhancers. Replacing Nhomie by an inverted version or by Homie (which pairs with itself head-to-head) transformed the stem loop into a circle loop. An important finding was that stem and circle loops differentially impact endogenous gene regulation both within the eve TAD and in the TADs bracketing eve. Intriguingly, an eve TAD with a circle loop configuration leads to ectopic activation of flanking genes by eve enhancers - indicating compromised regulatory boundary activity despite the presence of an eve TAD with intact left and right boundaries.

The results obtained are of high-quality and are meticulously discussed. This work advances our fundamental understanding of how 3D genome topologies affect enhancer-promoter communication.

This study raises interesting questions to be addressed in future studies.

First, given the unique specificity with which Nhomie and Homie pair (and exhibit "homing" activity), the generalizability of TAD formation by directional boundary pairing remains unclear. Testing whether boundary pairing is a phenomenon restricted to exceptional loci picked for study, rather than a broader rule of TAD formation, would best be done through the development of untargeted approaches to study boundary pairing.

Second, boundary pairing is one of several mechanisms that may form chromosomal contact domains such as TADs. Other mechanisms include cohesin-mediated chromosomal loop extrusion and the inherent tendency of transcriptionally active and inactive chromatin to segregate (or compartmentalize). The functional interplay between these possible TAD-forming mechanisms remains to be further investigated.

---

## [Referee Report · Reviewer #2 (Public Review)]

This study reports a set of experiments and subsequent analyses focusing on the role of *Drosophila* boundary elements in shaping 3D genome structure and regulating gene expression. The authors primarily focus on the region of the fly genome containing the even skipped (eve) gene; eve is expressed in a canonical spatial pattern in fly embryos and its locus is flanked by the well-characterized neighbor of homie (nhomie) and homie boundary elements. The main focus of the investigation is the orientation dependence of these boundary elements, which had been observed previously using reporter assays. In this study, the authors use Crispr/Cas9 editing followed by recombination-mediated cassette exchange to create a series of recombinant fly lines in which the nhomie boundary element is either replaced with exongenous sequence from phage 𝝀, an inversion of nhomie, or a copy of homie that has the same orientation as the endogenous homie sequence. The nhomie sequence is also regenerated in its native orientation to control for effects introduced by the transgenesis process.

The authors then perform high-resolution Micro-C to analyze 3D structure and couple this with fluorescent and colorimetric RNA in situ hybridization experiments to measure the expression of eve and nearby genes during different stages of fly development. The major findings of these experiments are that total loss of boundary sequence (replacement with 𝝀 DNA) results in major 3D structure changes and the most prominent observed gene changes, while inversion of the nhomie boundary or replacement with homie resulted in more modest effects in terms of 3D structure and gene expression changes and a distinct pattern of gene expression change from the 𝝀 DNA replacement. As the samples in which the nhomie boundary is inverted or replaced with homie have similar Micro-C profiles at the eve locus and show similar patterns of a spurious gene activation relative to the control, the observed effects appear to be driven by the relative orientation of the nhomie and homie boundary elements to one another.

Collectively, the findings reported in the manuscript are of broad interest to the 3D genome field. Although extensive work has gone into characterizing the patterns of 3D genome organization in a whole host of species, the underlying mechanisms that structure genomes and their functional consequences are still poorly understood. The perhaps best understood system, mechanistically, is the coordinated action of CTCF with the cohesin complex, which in vertebrates appears to shape 3D contact maps through a loop extrusion-pausing mechanism that relies on orientation-dependent sequence elements found at the boundaries of interacting chromatin loops. Despite having a CTCF paralog and cohesin, the *Drosophila* genome does not appear to be structured by loop extrusion-pausing. The identification of orientation-dependent elements with pronounced structural effects on genome folding thus may shed light on alternative mechanisms used to regulated genome structure, which in turn may yield insights into the significance of particular folding patterns.

On the whole, this study is comprehensive and represents a useful contribution to the 3D genome field. The transgenic lines and Micro-C datasets generated in the course of the work will be valuable resources for the research community. Moreover, the manuscript, while dense in places, is generally clearly written and comprehensive in its description of the work. However, I have a number of comments and critiques of the manuscript, mainly centering on the framing of the experiments and presentation of the Micro-C results and on the manner in which the data are analyzed and reported.

As this document now reflects my review of a revised version of the initial preprint, I will begin to add the new content at this point. As discussed in detail in the following paragraphs, my initial impression of the manuscript has not changed, so I have accordingly left the above text unaltered.

In my initial review, I provided a number of suggestions to improve the quality of the manuscript. These suggestions, which took the form of six major and three minor points, largely focused on (1) altering the writing in certain places to make the story more broadly accessible to the readership and (2) the inclusion of key, missing methodological detail to increase the rigor and reproducibility of the study. No new experiments were requested, and all of the points could be readily addressed with rather straightforward textual changes.

In their revised manuscript, the authors elected to directly address one of the major points and two of the minor points (major point 4, minor points 1 and 3). The remainder of my suggestions remain entirely unaddressed. A similar level of responsiveness was afforded to the very reasonable critiques of the other Reviewer and the Reviewing Editor. The authors have instead largely chosen to respond to the points raised exclusively in the rebuttal document. This document sprawls across >22 pages, includes numerous in-line figures, and cites dozens of references. The tone of this document, in many places, is at best forceful. In a less generous interpretation, many sections are combative, dismissive, and borderline unprofessional.

It is my opinion that the authors are doing the scientific community a disservice with their response. While it is my understanding that readers will be able see the rebuttal letter, I find that end result far from satisfying. How many readers will take the trouble to access that file, versus the manuscript itself? Skirting the review critiques places an unfair burden on readers, who are expecting peer-reviewed science, to dig into the accessory files to follow the critique and response, rather than seeing in reflected in the final product as they accustomed. Intentionally or not, the tactics the authors have chosen detract from what is otherwise a novel and well-intentioned new publishing model. It is also worth pointing out that peer review is done as an act of service to the scientific community, as the senior authors are doubtless aware. The other reviewer, the Reviewing Editor, and I have all taken time away from advancing our own careers and those of our trainees to offer the thoughtful critiques that were so pointedly dismissed.

In summary, as the vast majority of my critiques remain unaddressed, I have simply reproduced them below.

Major Points:

(1) The authors motivate much of the introduction and results with hypothetical "stem loop" and "circle loop" models of chromosome confirmation, which they argue are reflected in the Micro-C data and help to explain the observed ISH patterns. While such structures may possibly form, the support for these specific models vs. the many alternatives is not in any way justified. For instance, no consideration is given to important biophysical properties such as persistence length, packing/scaling, and conformational entropy. As the biophysical properties of chromatin are a very trafficked topic both in terms of experimentation and computational modeling and generally considered in the analysis of chromosome conformation data, the study would be strengthened by acknowledgement of this body of work and more direct integration of its findings.

(2) Similar to Point 1, while there is a fair amount of discussion of how the observed results are or are not consistent with loop extrusion, there is no discussion of the biophysical forces that are thought to underly compartmentalization such as block-polymer co-segregation and their potential influence. I found this absence surprising, as it is generally accepted that A/B compartmentalization essentially can explain the contact maps observed in *Drosophila* and other non-vertebrate eukaryotes (Rowley, ..., Corces 2017; PMID 28826674). The manuscript would be strengthened by consideration of this phenomenon.

(3) The contact maps presented in the study represent many cells and distinct cell types. It is clear from single-cell Hi-C and multiplexed FISH experiments that chromosome conformation is highly variable even within populations of the same cell, let alone between cell types, with structures such as TADs being entirely absent at the single cell level and only appearing upon pseudobulking. It is difficult to square these observations with the models of relatively static structures depicted here. The authors should provide commentary on this point.

(4) Related to Point 4, the lack of quantitative details about the Micro-C data make it difficult to evaluate if the changes observed are due to biological or technical factors. It is essential that the authors provide quantitative means of controlling for factors like sampling depth, normalization, and data quality between the samples.

(5) The ISH effects reported are modest, especially in the case of the HCR. The details provided for how the imaging data were acquired and analyzed are minimal, which makes evaluating them challenging. It would strengthen the study to provide much more detail about the acquisition and analysis and to include depiction of intermediates in the analysis process, e.g. the showing segmentation of stripes.

---

## [Author Response]

The following is the authors’ response to the original reviews.

**Reviewer #1 (Public Review):**
Summary:In this study, the authors engineer the endogenous left boundary of the *Drosophila* eve TAD, replacing the endogenous Nhomie boundary by either a neutral DNA, a wildtype Nhomie boundary, an inverted Nhomie boundary, or a second copy of the Homie boundary. They perform Micro-C on young embryos and conclude that endogenous Nhomie and Homie boundaries flanking eve pair with head-to-tail directionality to form a chromosomal stem loop. Abrogating the Nhomie boundary leads to ectopic activation of genes in the former neighboring TAD by eve embryonic stripe enhancers. Replacing Nhomie by an inverted version or by Homie (which pairs with itself head-to-head) transformed the stem loop into a circle loop. An important finding was that stem and circle loops differentially impact endogenous gene regulation both within the eve TAD and in the TADs bracketing eve. Intriguingly, an eve TAD with a circle loop configuration leads to ectopic activation of flanking genes by eve enhancers - indicating compromised regulatory boundary activity despite the presence of an eve TAD with intact left and right boundaries.Strengths:Overall, the results obtained are of high-quality and are meticulously discussed. This work advances our fundamental understanding of how 3D genome topologies affect enhancer-promoter communication.Weaknesses:Though convincingly demonstrated at eve, the generalizability of TAD formation by directional boundary pairing remains unclear, though the authors propose this mechanism could underly the formation of all TADs in *Drosophila* and possibly even in mammals. Strong and ample evidence has been obtained to date that cohesin-mediated chromosomal loop extrusion explains the formation of a large fraction of TADs in mammals.

**(1.1)** The difficulty with most all of the studies on mammal TADs, cohesin, and CTCF roadblocks is that the sequencing depth is not sufficient, and large bin sizes (>1 kb) are needed to visualize chromosome architecture. The resulting contact profiles show TAD neighborhoods, not actual TADs.

This problem is illustrated by comparing the contact profiles of mammalian MicroC data sets at different bin sizes in Author response image 1. In this figure, the darkness of the “pixels” in panels E, F, G, and H was enhanced by reducing brightness using Photoshop.

**Author response image 1. sa3fig1:** Mammalian MicroC profiles at different bin sizes.

[Author response image 1 is adapted from Krietenstein et al., 2020; Hsieh et al., 2020.]

Panels A and C are from Krietenstein et al. (2020), and show “TADs” using bin sizes typical of most mammalian studies. At this level of resolution, TADs, the “trees” that are the building blocks of chromosomes, are not visible. Instead, what is seen are TAD neighborhoods or “forests”. Each neighborhood consists of several dozen individual TADs. The large bins in these panels also artificially accentuated TAD:TAD interactions, generating a series of “stripes” and “dots” that correspond to TADs bumping into each other and sequences getting crosslinked. For example, in panel A there is a prominent stripe on the edge of a “TAD” (blue arrow). In panel C, this stripe resolves into a series of dots arranged as parallel, but interrupted, “stripes” (green and blue arrows). At the next level of resolution, it can be seen that the stripe marked by the blue arrow and magenta asterisk is generated by contacts between the left boundary of the TAD indicated by the magenta bar with sequences in a TAD (blue bar) ~180 kb way. While dots and stripes are prominent features in contact profiles visualized with larger bin sizes (A and C), the actual TADs that are observed with a bin size of 200 bp (examples are underlined by black bars in panel G) are not bordered by stripes, nor are they topped by obvious dots. The one possible exception is the dot that appears at the top of the volcano triangle underlined with magenta.

The chromosome 1 DNA segment from the MicroC data of Hsieh et al. (2020) shows a putative volcano triangle with a plume (indicated by a V in Author response image 1, panels D, F, and H). Sequences in the V TAD don’t crosslink with their immediate neighbors, and this gives a “plume” above the volcano triangle, as indicate by the light blue asterisk in panels D, F, and H. Interestingly, the V TAD does contact two distant TADs, U on the left and W on the right. The U TAD is ~550 kb from V, and the region of contact is indicated by the black arrow. The W TAD is ~585 kb from V, and the region of contact is indicated by the magenta arrow. While the plume still seems to be visible with a bin size of 400 bp (light blue asterisk), it is hard to discern when the bin size is 200 bp, as there are not enough reads.

The evidence demonstrating that cohesin is required for TAD formation/maintenance is based on low resolution Hi-C data, and the effects that are observed are on TAD neighborhoods (forests), and not TADs (trees). In fact, there is published evidence that cohesin is not required in mammals for TAD formation/maintenance. Author response image 2 shows the *Ppm1g* region of mouse chromosome 5 generated from data in a paper by Goel et al. (2023). In this experiment, the authors depleted the cohesin component RAD21 and then visualized the effects on TAD organization using the high resolution region capture MicroC (RCMC) protocol. The MicroC contact map in Author response image 2 visualizes a ~150 kb DNA segment around the *Ppm1pg* locus at 250 bp resolution. On the right side of the diagonal is the untreated control, while the left side shows the MicroC profile of the same region after RAD21 depletion. The authors indicated that there was a 97% depletion of RAD21 in their experiment. However, as is evident from a comparison of the experimental and control, loss of RAD21 has no apparent effect on the TAD organization of this mammalian DNA segment. Likewise, TAD:TAD interactions between next door neighbors (purple asterisks), next-next door neighbors (red asterisks) and next-next-next door neighbors (blue asterisks) is not perturbed either.

**Author response image 2. sa3fig2:** 

[Author response image 2 is generated from data available from Goel et al., 2023.]

Several other features are worth noting. First, unlike the MicroC experiments shown in Author response image 1, there are dots at the apex of the TADs in this chromosomal segment. In the MicroC protocol, fixed chromatin is digested to mononucleosomes by extensive MNase digestion. The resulting DNA fragments are then ligated, and dinucleosome-length fragments are isolated and sequenced. DNA sequences that are nucleosome free in chromatin (which would be promoters, enhancers, silencers and boundary elements) are typically digested to oligonucleotides in this procedure and won’t be recovered. This means that the dots shown here must correspond to mononucleosome-length elements that are MNase resistant. This is also true for the dots in the MicroC contact profiles of the *Drosophila Abd-B* regulatory domain (see Fig. 2B in the paper). Second, the TADs are connected to each other by 45^o^ stripes (see blue and green arrowheads). While it is not clear from this experiment whether the stipes are generated by an active mechanism (enzyme) or by some “passive” mechanism (e.g., sliding), the stripes in this chromosomal segment *are not* generated by cohesin, as they are unperturbed by RAD21 depletion. Third, there are no volcano triangles with plumes in this chromosomal DNA segment. Instead, the contact patterns between neighboring TADs closely resemble those seen for the *Abd-B* regulatory domains (compare Author response image 2 with Fig. 2B in the paper). This similarity suggests that the TADs in and around *Ppm1g* may be circle-loops, not stem-loops. As volcano triangles with plumes also seem to be rare in the MicroC data sets of Krietenstein et al. (2020) and Hsieh et al. (2020) (with the caveat that these data sets are low resolution: see Author response image 1), it is possible that much of the mammalian genome is assembled into circle-loop TADs, a topology that can’t be generated by the cohesin loop-extrusion (bolo tie clip) /CTCF roadblock model.

**Author response image 3. sa3fig3:** 

[Author response image 3 is generated from data available from Goel et al., 2023.]

While RAD21 depletion has no apparent effect on TADs, it does appear to have a modest impact on TAD neighborhoods. This was shown in a supplemental figure in Goel et al. (2023), which visualized the *Ppm1g* region of chromosome 5 with bin sizes of 5 kb and 1 kb. Author response image 3 shows an ~600 kb region from chromosome 5 containing the *Ppm1g* gene, visualized with a bin size of 1 kb. As can be seen from comparing the MicroC profiles in this image with that in Author response image 2, individual TADs are not visible. Instead, the individual TADs are binned into large TAD “neighborhoods” that consist of multiple TADs.

Unlike the individual TADs shown in Author response image 2, the TAD neighborhoods in Author response image 3 show a limited sensitivity to RAD21 depletion. The effects of RAD21 depletion can be seen by comparing the relative pixel density in the box before (blue box above the diagonal) and after auxin-induced RAD21 degradation (purple box below the diagonal). The reduction in pixel density is greatest for more distant TAD:TAD contacts (farthest from the diagonal: green double arrow). By contrast, the TADs themselves are unaffected, as are contacts between individual TADs and their neighbors (Author response image 2 above). A subset of higher density contact “dots” (green asterisks) also appear to be reduced after RAD21 depletion, though the effects are not uniform (blue asterisks). At this point it isn’t clear why contacts between distant TADs in the same neighborhood are lost when RAD21 is depleted; however, a plausible speculation is that it is related to the functioning of cohesin in holding newly replicated DNAs together until mitosis, and whatever other role(s) it might have in chromosome condensation.

Moreover, given the unique specificity with which Nhomie and Homie are known to pair (and exhibit "homing" activity), it is conceivable that formation of the eve TAD by boundary pairing represents a phenomenon observed at exceptional loci rather than a universal rule of TAD formation. Indeed, characteristic Micro-C features of the eve TAD are only observed at a restricted number of loci in the fly genome…..

**(1.2)** The available evidence does not support the claim that *nhomie* and *homie* are “exceptional.” To begin with, *nhomie* and *homie* rely on precisely the same set of factors that have been implicated in the functioning of other boundaries in the fly genome. For example, *homie* requires (among other factors) the generic boundary protein Su(Hw) for insulation and long-distance interactions (Fujioka et al. 2024). (This is also true of *nhomie*: unpublished data.) The Su(Hw) protein (like other fly polydactyl zinc finger proteins) can engage in distant interactions. This was first shown by Sigrist and Pirrotta (Sigrist and Pirrotta 1997), who found that the *su(Hw)* element from the *gypsy* transposon can mediate long-distance regulatory interactions (PRE-dependent silencing) between transgenes inserted at different sites on homologous chromosomes (*trans* interactions) and at sites on different chromosomes.

The ability to mediate long-distance interactions is not unique to the *su(Hw)* element, or *homie* and *nhomie*. Muller et al. (1999) found that the *Mcp* boundary from the *Drosophila* BX-C is also able to engage in long-distance regulatory interactions: both PRE-dependent silencing of *mini-white* and enhancer activation of *mini-white* and *yellow*. The functioning of the *Mcp* boundary depends upon two other generic insulator proteins, Pita and the fly CTCF homolog (Kyrchanova et al. 2017). Like Su(Hw), both are polydactyl zinc finger proteins, and they resemble the mammalian CTCF protein in that their N-terminal domain mediates multimerization (Bonchuk et al. 2020; Zolotarev et al. 2016). Author response image 4 shows PRE-dependent “pairing sensitive silencing” interactions between transgenes carrying a *mini-white* reporter, the *Mcp* and *scs’* (BEAF-dependent, Hart et al. 1997) boundary elements, and a PRE closely linked to *Mcp*. In this experiment, flies homozygous for different transgene inserts were mated and the eye color was examined in their *trans-*heterozygous progeny. As indicated in the figure, the strongest *trans-*silencing interactions were observed for inserts on the same chromosomal arm; however, transgenes inserted on the left arm of chromosome 3 can interact across the centromere with transgenes inserted on the right arm of chromosome 3.

**Author response image 4. sa3fig4:** 

[Author response image 4 is reproduced from Figure 6 from Muller et al., 1999, with permission from Genetics Society of America. It is not covered by the CC-BY 4.0 license, and further reproduction of this figure would need permission from the copyright holder.]

Author response image 5A shows a *trans*-silencing interaction between *w#11.102* at 84D and *w#11.16* approximately 5.8 Mb away, at 87D. Author response image 5B shows a *trans*-silencing interaction across the centromere between *w#14.29* on the left arm of chromosome 3 at 78F and *w#11.102* on the right arm of chromosome 3 at 84D. The eye color phenotype of *mini-white*-containing transgenes is usually additive: homozygyous inserts have twice as dark eye color as the corresponding hemizygous inserts. Likewise, in flies *trans-*heterozygous for *mini-white* transgenes inserted at different sites, the eye color is equivalent to the sum of the two transgenes. This is not true when *mini-white* transgenes are silenced by PREs. In the combination shown in panel A, the *trans-*heterozygous fly has a lighter eye color than either of the parents. In the combination in panel B, the *trans-*heterozygous fly is slightly lighter than either parent.

**Author response image 5. sa3fig5:** Long-distance pairing-sensitive silencing between transgenes inserted on 3R. Transgene insertion sites are shown in Author response image 4.

[Author response image 5 is reproduced from Figure 5C from Muller et al., 1999, with permission from Genetics Society of America. It is not covered by the CC-BY 4.0 license, and further reproduction of this figure would need permission from the copyright holder.]

As evident from the diagram in Author response image 4, all of the transgenes inserted on the 3^rd^ chromosome that were tested were able to participate in long-distance (>Mbs) regulatory interactions. On the other hand, not all possible pairwise interactions are observed. This would suggest that potential interactions depend upon the large scale (Mb) 3D folding of the 3^rd^ chromosome.

When the *scs* boundary (Zw5-dependent, Gaszner et al. 1999) was added to the transgene to give *sMws’*, it further enhanced the ability of distant transgenes to find each other and pair. All eight of the *sMws’* inserts that were tested were able to interact with at least one other *sMws’* insert on a *different* chromosome and silence *mini-white*. Vazquez et al. (2006) subsequently tagged the *sMws’* transgene with LacO sequences (*ps0Mws’*) and visualized pairing interactions in imaginal discs. *Trans*-heterozygous combinations on the same chromosome were found paired in 94-99% of the disc nuclei, while a *trans*-heterozygous combination on different chromosomes was found paired in 96% of the nuclei (Author response image 6). Vazquez et al. (2006) also examined a combination of four transgenes inserted on the same chromosome (two at the same insertion site, and two at different insertion sites). In this case, all four transgenes were clustered together in 94% of the nuclei (Author response image 6). Their studies also suggest that the distant transgenes remain paired for at least several hours. A similar experiment was done by Li et al. (2011), except that the transgene contained only a single boundary, *Mcp* or *Fab-7*. While pairing was still observed in *trans*-heterozygotes, the frequency was reduced without *scs* and *scs’*.

**Author response image 6. sa3fig6:** 

[Author response image 6 is reproduced from Table 3 from Vazquez et al., 2006.]

It is worth pointing out that there is no plausible mechanism in which cohesin could extrude a loop through hundreds of intervening TADs, across the centromere (*ff#13.101* with *w#11.102*: Author response image 4; *w#14.29 with w#11.02*: Author response images 4 and 5) and come to a halt when it “encounters” *Mcp*-containing transgenes on different homologs. The same is true for *Mcp*-dependent pairing interactions in *cis* (Fig. 7 in Muller et al. 1999) or *Mcp*-dependent pairing interactions between transgenes inserted on different chromosomes (Line 8 in Author Response Figure 6, Fig. 8 in Muller et al. 1999).

These are not the only boundaries that can engage in long-distance pairing. Mohana et al. (2023) identified nearly 60 meta-loops, many of which appear to be formed by the pairing of TAD boundary elements. Two examples (at 200 bp resolution from 12-16 hr embryos) are shown in Author response image 7.

**Author response image 7. sa3fig7:** Metaloops on the 2^nd^ and 3^rd^ chromosomes: circle-loops and multiple stem-loops.

One of these meta-loops (panel A) is generated by the pairing of two TAD boundaries on the 2^nd^ chromosome. The first boundary, blue (indicated by blue arrow), is located at ~2,006,500 bp, between a small TAD containing the *Nplp4* and *CG15353* genes and a larger TAD containing 3 genes, *CG33543*, *Obp22a,* and *Npc2a*. *Nplp4* encodes a neuropeptide. The functions of *CG15354* and *CG33543* are unknown. *Obp22a* encodes an odorant binding protein, while *Npc2a* encodes the Niemann-Pick type C-2a protein that is involved sterol homeostasis. The other boundary (purple: indicated by purple arrow) is located between two TADs 2.8 Mb away at 4,794,250 bp. The upstream TAD contains the *fipi* gene (*CG15630*) which has neuronal functions in male courtship, while the downstream TAD contains *CG3294*, which is thought to be a spliceosome component, and *schlaff* (*slf*), which encodes a chitin binding protein. As illustrated in the accompanying diagram, the blue boundary pairs with the purple boundary in a head-to-head orientation, generating a ~2.8 Mb loop with a circle-loop topology. As a result of this pairing, the multi-gene (*CG33543*, *Obp22a,* and *Npc2a*) TAD upstream of the blue boundary interacts with the *CG15630* TAD upstream of the purple boundary. Conversely the small *Nplp4:CG15353* TAD downstream of the blue boundary interacts with the *CG3294:slf* TAD downstream of the purple boundary. Even if one imagined that the cohesin bolo tie clip was somehow able to extrude 2.8 Mb of chromatin and then know to stop when it encountered the blue and purple boundaries, it would’ve generated a stem-loop, not a circle-loop.

The second meta-loop (panel B) is more complicated, as it involves pairing interactions between four boundary elements. The blue boundary (blue arrow), located ~4,801,800 bp (3L), separates a large TAD containing the *RhoGEF64C* gene from a small TAD containing *CG7509,* which encodes a predicted subunit of an extracellular carboxypeptidase. As can be seen in the MicroC contact profile and the accompanying diagram, the blue boundary pairs with the purple boundary (purple arrow), which is located at ~7,013, 500 (3L), just upstream of the 2^nd^ internal promoter (indicated by black arrowhead) of the *Mp* (*Multiplexin*) gene. This pairing interaction is head-to-tail and generates a large stem-loop that spans ~2.2 Mb. The stem-loop brings sequences upstream of the blue boundary and downstream of the purple boundary into contact (like the strings below a bolo tie clip), just as was observed in the boundary bypass experiments of Muravyova et al. (2001) and Kyrchanova et al. (2008). The physical interactions result in a box of contacts (right top) between sequences in the large *RhoGEF64C* TAD and sequences in a large TAD that contains an internal *Mp* promoter. The second pairing interaction is between the brown boundary (brown arrow) and the green boundary (green arrow). The brown boundary is located at ~4 805,600 bp (3L), and separates the TAD containing *CG7590* from a large TAD containing *CG1808* (predicted to encode an oxidoreductase) and the *Dhc64C* (*Dynein heavy chain 64C*) gene. The green boundary is located at ~6,995,500 bp (3L), and it separates a TAD containing *CG32388* and the *biniou* (*bin*) transcription factor from a TAD that contains the most distal promoter of the *Mp* gene (blue arrowhead). As indicated in the diagram, the brown and green boundaries pair with each other head-to-tail, and this generates a small internal loop (and the final configuration would resemble a bolo tie with two tie clips). This small internal loop brings the *CG7590* TAD into contact with the TAD that extends from the distal *Mp* promoter to the 2^nd^ internal *Mp* promoter. The resulting contact profile is a rectangular box with diagonal endpoints corresponding to the paired blue:purple and brown:green boundaries. The pairing of the brown:green boundaries also brings the TADs immediately downstream of the brown boundary and upstream of the green boundary into contact with each other, and this gives a rectangular box of interactions between the *Dhc64C* TAD and sequences in the *bin*/*CG3238* TAD. This box is located on the lower left side of the contact map.

Since the *bin* and *Mp* meta-loops in Author response image 7B are stem-loops, they could have been generated by “sequential” cohesin loop extrusion events. Besides the fact that cohesin extrusion of 2 Mb of chromatin and breaking through multiple intervening TAD boundaries challenges the imagination, there is no mechanism in the cohesion loop extrusion/CTCF roadblock model to explain why cohesion complex 1 would come to a halt at the purple boundary on one side and the blue boundary on the other, while cohesin complex 2 would instead stop when it hits the brown and green boundaries. This highlights another problem with the cohesin loop extrusion/CTCF roadblock model, namely that the roadblocks are *functionally autonomous*: they have an intrinsic ability to block cohesin that is entirely independent of the intrinsic ability of other roadblocks in the neighborhood. As a result, there is no mechanism for generating specificity in loop formation. By contrast, boundary pairing interactions are by definition non-autonomous and depend on the ability of individual boundaries to pair with other boundaries: specificity is built into the model.

The mechanism for pairing, and accordingly the basis for partner preferences/specificity, are reasonably well understood. Probably the most common mechanism in flies is based on shared binding sites for architectural proteins that can form dimers or multimers (Bonchuk et al. 2021; Fedotova et al. 2017). Flies have a large family of polydactyl zinc finger DNA binding proteins, and as noted above, many of these form dimers or multimers, and also function as TAD boundary proteins. This pairing principle was first discovered by Kyrchanova et al. (2008). This paper also showed that orientation-dependent pairing interactions is a common feature of endogenous fly boundaries. Another mechanism for pairing is specific protein:protein interactions between different DNA binding factors (Blanton et al. 2003). Yet a third mechanism would be proteins that bridge different DNA binding proteins together. The boundaries that use these different mechanisms (BX-C boundaries, *scs*, *scs’*) depend upon the same sorts of proteins that are used by *homie* and *nhomie*. Likewise, this same set of factors reappears, in one combination or another, in most other TAD boundaries. As for the orientation of pairing interactions, this is most likely determined by the order of binding sites for chromosome architectural proteins in the partner boundaries.

…. and many TADs lack focal 3D interactions between their boundaries.

**(1.3)** The evidence that flies differ from mammals in that they “lack” focal 3D interactions is not compelling. One of the problems with drawing this distinction is that almost all of the “focal 3D interactions” seen mammalian Hi-C experiments are a consequence of binning large DNA segments in low resolution restriction enzyme-dependent experiments. This is even true in the two “high” resolution MicroC experiments that have been published (Hsieh et al. 2020; Krietenstein et al. 2020). As illustrated above in Author response image 1, most of the “focal 3D interactions” (the dots at the apex of TAD triangles) seen with large bin sizes (1 kb and greater) disappear when the bin size is 200 bp, and TADs rather than TAD neighborhoods are being visualized.

As described in point (1.1) above, in the MicroC protocol, fixed chromatin is first digested to mononucleosomes by extensive MNase digestion, processed/biotinylated, and ligated to give dinucleosome-length fragments, which are then sequenced. Regions of chromatin that are nucleosome free (promoters, enhancers, silencers, boundary elements) will typically be reduced to oligonucleotides in this procedure and will not be recovered when dinucleosome-length fragments are sequenced. The loss of sequences from typical paired boundary elements is illustrated by the *Lar* (*Leukocyte-antigen-related-like*) meta-loop shown in Author response image 8 (at 200 bp resolution). Panels A and B show the contact profiles generated when the blue boundary (which separates two TADs that span the *Lar* transcription unit interacts with the purple boundary (which separates two TADs in a gene poor region ~620 kb away)). The blue and purple boundaries pair with each other head-to-head, and this pairing orientation generates yet another circle-loop. In the circle-loop topology, sequences in the TADs upstream of both boundaries come into contact with each other, and this gives the small dark rectangular box to the upper left of the paired boundaries (Author response image 8A). (Note that this small box corresponds to the two small TADs upstream of the blue and purple boundaries, respectively. See panel B.) Sequences in the TADs downstream of the two boundaries also come into contact with each other, and this gives the large box to the lower right of the paired boundaries. While this meta-loop is clearly generated by pairing interactions between the blue and purple boundaries, the interacting sequences are degraded in the MicroC protocol, and sequences corresponding to the blue and purple boundaries aren’t recovered. This can be seen in panel B (red arrow and red arrowheads). When a different Hi-C procedure is used (dHS-C) that captures nucleosome-free regions of chromatin that are physically linked to each other (Author response image 8C, D), the sequences in the interacting blue and purple boundaries are recovered and generate a prominent “dot” at their physical intersection (blue arrow in panel D).

**Author response image 8. sa3fig8:** *Lar* metaloop. (A, B) MicroC. (C, D) dHS-C.

While sequences corresponding to the blue and purple boundaries are lost in the MicroC procedure, there is at least one class of element that engages in physical pairing interactions whose sequences are (comparatively) resistant to MNase digestion. This class of elements includes many PREs (Kyrchanova et al. 2018; unpublished data), the boundary bypass elements in the *Abd-B* region of the BX-C (Kyrchanova et al. 2023; Kyrchanova et al. 2019a; Kyrchanova et al. 2019b; Postika et al. 2018), and “tethering” elements (Batut et al. 2022; Li et al. 2023). In all of the cases tested, these elements are bound in nuclear extracts by a large (>1000 kD) GAGA factor-containing multiprotein complex called LBC. LBC also binds to the *hsp70* and *eve* promoters (unpublished data). Indirect end-labeling experiments (Galloni et al. 1993; Samal et al. 1981; Udvardy and Schedl 1984) indicate that the LBC protects a ~120-180 bp DNA segment from MNase digestion. It is likely that this is the reason why LBC-bound sequences can be recovered in MicroC experiments as dots when they are physically linked to each other. One such example (based on the ChIP signatures of the paired elements) is indicated by the green arrow in panel B and D of Author response image 8. Note that there are no dots corresponding to these two LBC elements within either of the TADs immediately downstream of the blue and purple boundaries. Instead the sequences corresponding to the two LBC elements are only recovered when the two elements pair with each other over a distance of ~620 kb. The fact that these two elements pair with each other is consistent with other findings which indicate that, like classical boundaries, LBC elements exhibit partner preferences. In fact, LBC elements can sometimes function as TAD boundaries. For example, the *Fab-7* boundary has two LBC elements, and full *Fab-7* boundary function can be reconstituted with just these two elements (Kyrchanova et al. 2018).

**Reviewer #2 (Public Review):**
"Chromatin Structure II: Stem-loops and circle-loops" by Ke*, Fujioka*, Schedl, and Jaynes reports a set of experiments and subsequent analyses focusing on the role of *Drosophila* boundary elements in shaping 3D genome structure and regulating gene expression. The authors primarily focus on the region of the fly genome containing the even skipped (eve) gene; eve is expressed in a canonical spatial pattern in fly embryos and its locus is flanked by the well-characterized neighbor of homie (nhomie) and homie boundary elements. The main focus of investigation is the orientation dependence of these boundary elements, which had been observed previously using reporter assays. In this study, the authors use Crispr/Cas9 editing followed by recombination-mediated cassette exchange to create a series of recombinant fly lines in which the nhomie boundary element is either replaced with exongenous sequence from phage 𝝀, an inversion of nhomie, or a copy of homie that has the same orientation as the endogenous homie sequence. The nhomie sequence is also regenerated in its native orientation to control for effects introduced by the transgenesis process.The authors then perform high-resolution Micro-C to analyze 3D structure and couple this with fluorescent and colorimetric RNA in situ hybridization experiments to measure the expression of eve and nearby genes during different stages of fly development. The major findings of these experiments are that total loss of boundary sequence (replacement with 𝝀 DNA) results in major 3D structure changes and the most prominent observed gene changes, while inversion of the nhomie boundary or replacement with homie resulted in more modest effects in terms of 3D structure and gene expression changes and a distinct pattern of gene expression change from the 𝝀 DNA replacement. As the samples in which the nhomie boundary is inverted or replaced with homie have similar Micro-C profiles at the eve locus and show similar patterns of a spurious gene activation relative to the control, the observed effects appear to be driven by the relative orientation of the nhomie and homie boundary elements to one another.Collectively, the findings reported in the manuscript are of broad interest to the 3D genome field. Although extensive work has gone into characterizing the patterns of 3D genome organization in a whole host of species, the underlying mechanisms that structure genomes and their functional consequences are still poorly understood. The perhaps best understood system, mechanistically, is the coordinated action of CTCF with the cohesin complex, which in vertebrates appears to shape 3D contact maps through a loop extrusion-pausing mechanism that relies on orientation-dependent sequence elements found at the boundaries of interacting chromatin loops.

**(2.1)** The notion that the mammalian genome is shaped in 3D by the coordinate action of cohesin and CTCF has achieved the status of dogma in the field of chromosome structure in vertebrates. However, as we have pointed out in (1.1), the evidence supporting this dogma is far from convincing. To begin with, it is based on low resolution Hi-C experiments that rely on large bin sizes to visualize so-called “TADs.” In fact, the notion that cohesin and CTCF are responsible on their own for shaping the mammalian 3D genome appears to be a result of mistaking a series of forests for the actual trees that populate each of the forests.

As illustrated in Author response image 1 above, the “TADs” that are visualized in these low resolution data sets are not TADs at all, but rather TAD neighborhoods consisting of several dozen or more individual TADs. Moreover, the “interesting” features that are evident at low resolution (>1 kb), the dots and stripes, largely disappear at resolutions appropriate for visualizing individual TADs (~200 bp).

In Author response image 2, we presented data from one of the key experiments in Goel et al. (2023). In their experiment, the authors used RCMC to generate high resolution MicroC contact maps before and after RAD21 depletion. Contrary to dogma, RAD21 depletion has *no effect* on TADs in a chromosome 5 DNA segment spanning the *Ppm1g* gene, when TADs are visualized at 200 bp resolution (as in Author response image 2) or at 250 bp resolution (as in Goel et al. 2023). These TADs look very much like the TADs we observe in the *Drosophila* genome, in particular, in the *Abd-B* region of the BX-C that is thought to be assembled into a series of circle-loops (see Fig. 2B).

While Goel et al. (2023) observed no effect of RAD21 depletion on TADs, they found that loss of RAD21 does have some impact on the frequency of longer-distance (but not short-distance) contacts in TAD neighborhoods when their RCMC data set is visualized using bin sizes of 5 kb and 1 kb. This is shown using data from their paper, imaged using a bin size of 1 kb, in Author response image 3. The significance of this finding is, however, uncertain. It could mean that the 3D organization of large TAD neighborhoods have a special requirement for cohesin activity. On the other hand, since cohesin functions to hold sister chromosomes together after replication until they separate during mitosis (and might also participate in mitotic condensation), it is also possible that the loss of long-range contacts in large TAD neighborhoods when RAD21 is depleted is simply a reflection of this particular activity. Further studies will be required to address these possibilities.

As for CTCF: a careful inspection of the ChIP data in Author response image 2 indicates that CTCF is *not* found at each and every TAD boundary. In fact, the notion that CTCF is the *be-all and end-all* of TAD boundaries in mammals is truly hard to fathom. For one, the demands for specificity in TAD formation (and in regulatory interactions) are likely much greater than those in flies, and specificity can’t be generated by a single DNA binding protein. For another, several dozen chromosomal architectural proteins have already been identified in flies. This means that (unlike what is *thought* to be true in mammals) it is possible to use a combinatorial mechanism to generate specificity in, for example, the long-distance interactions in Author response images 7 and 8. As noted in (2.1) above, many of the known chromosomal architectural proteins in flies are polydactyl zinc finger proteins (just like CTCF). There are some 200 different polydactyl zinc finger proteins in flies, and the function of only a hand full of these is known at present. However, it seems likely that a reasonable fraction of this class of DNA binding proteins will ultimately turn out to have an architectural function of some type (Bonchuk et al. 2021; Fedotova et al. 2017). The number of different polydactyl zinc finger protein genes in mammals is nearly 3 times that of flies. It is really possible that of these, only CTCF is involved in shaping the 3D structure of the mammalian genome?

Despite having a CTCF paralog and cohesin, the *Drosophila* genome does not appear to be structure by loop extrusion-pausing. The identification of orientation-dependent elements with pronounced structural effects on genome folding thus may shed light on alternative mechanisms used to regulated genome structure, which in turn may yield insights into the significance of particular folding patterns.

**(2.2)** Here we would like to draw the reviewer’s and reader’s attention to Author response image 7, which shows that orientation-dependent pairing interactions have a significant impact on physical interactions between different sequences. We would also refer the reader to two other publications. One of these is Kyrchanova et al. (2008), which was the first to demonstrate that orientation of pairing interactions matters. The second is Fujioka et al. (2016), which describes experiments indicating that *nhomie* and *homie* pair with each other head-to-tail and with themselves head-to-head.

On the whole, this study is comprehensive and represents a useful contribution to the 3D genome field. The transgenic lines and Micro-C datasets generated in the course of the work will be valuable resources for the research community. Moreover, the manuscript, while dense in places, is generally clearly written and comprehensive in its description of the work. However, I have a number of comments and critiques of the manuscript, mainly centering on the framing of the experiments and presentation of the Micro-C results and on manner in which the data are analyzed and reported. They are as follows:Major Points:(1) The authors motivate much of the introduction and results with hypothetical "stem loop" and "circle loop" models of chromosome confirmation, which they argue are reflected in the Micro-C data and help to explain the observed ISH patterns. While such structures may possibly form, the support for these specific models vs. the many alternatives is not in any way justified. For instance, no consideration is given to important biophysical properties such as persistence length, packing/scaling, and conformational entropy. As the biophysical properties of chromatin are a very trafficked topic both in terms of experimentation and computational modeling and generally considered in the analysis of chromosome conformation data, the study would be strengthened by acknowledgement of this body of work and more direct integration of its findings.

**(2.3)** The reviewer is not correct in claiming that “stem-loops” and “circle-loops” are “hypothetical.” There is ample evidence that both types of loops are present in eukaryotic genomes, and that loop conformation has significant readouts in terms of not only the physical properties of TADs but also their functional properties. Here we would draw the reviewer’s attention to Author response images 7 and 8 for examples of loops formed by the orientation-dependent pairing of yet other TAD boundary elements. As evident from the MicroC data in these figures, circle-loops and stem-loops have readily distinguishable contact patterns. The experiments in Fujioka et al. (2016) demonstrate that *homie* and *nhomie* pair with each other head-to-tail, while they pair with themselves head-to-head. The accompany paper (Bing et al. 2024) also provides evidence that loop topology is reflected both in the pattern of activation of reporters and in the MicroC contact profiles. We would also mention again Kyrchanova et al. (2008), who were the first to report orientation-dependent pairing of endogenous fly boundaries.

At this juncture, it would premature to try to incorporate computational modeling of chromosome conformation in our studies. The reason is that the experimental foundations that would be essential for building accurate models are lacking. As should be evident from Author response images 1-3 above, studies on mammalian chromosomes are simply not of high enough resolution to draw firm conclusions about chromosome conformation: in most studies only the forests are visible. While the situation is better in flies, there are still too many unknown. As just one example, it would be important to know the orientation of the boundary pairing interactions that generate each TAD. While it is possible to infer loop topology from how TADs interact with their neighbors (a plume versus clouds), a conclusive identification of stem- and circle-loops will require a method to unambiguously determine whether a TAD boundary pairs with its neighbor head-to-head or head-to-tail.

(2) Similar to Point 1, while there is a fair amount of discussion of how the observed results are or are not consistent with loop extrusion, there is no discussion of the biophysical forces that are thought to underly compartmentalization such as block-polymer co-segregation and their potential influence. I found this absence surprising, as it is generally accepted that A/B compartmentalization essentially can explain the contact maps observed in *Drosophila* and other non-vertebrate eukaryotes (Rowley, ..., Corces 2017; PMID 28826674). The manuscript would be strengthened by consideration of this phenomenon.

**(2.4)** As the reviewer indicates, an alternative mechanism for generating TADs and for explaining the patterns of TAD:TAD interactions is provided by the A/B compartment model. This model is not consistent with the experiments described in either this manuscript or in the accompanying manuscript (Bing et al. 2024). Nor does it fit with extensive studies on the structural and functional properties of boundary elements in the *Abd-B* region of the bithorax complex. As the reviewer has suggested, we have included a discussion of the A/B compartment model in the last section of the Discussion.

(3) The contact maps presented in the study represent many cells and distinct cell types. It is clear from single-cell Hi-C and multiplexed FISH experiments that chromosome conformation is highly variable even within populations of the same cell, let alone between cell types, with structures such as TADs being entirely absent at the single cell level and only appearing upon pseudobulking. It is difficult to square these observations with the models of relatively static structures depicted here. The authors should provide commentary on this point.

**(2.5)** As should be evident from Author response image 1, single-cell Hi-C experiments would not provide useful information about the physical organization of individual TADs, TAD boundaries, or how individual TADs interact with their immediate neighbors. In addition, since they capture only a very small fraction of the possible contacts within and between TADs, we suspect that these single-cell studies aren’t likely to be useful for making solid conclusions about TAD neighborhoods like those shown in Author response image 1 panels A, B, C, and D, or Author response image 3. While it might be possible to discern relatively stable contacts between pairs of insulators in single cells with the right experimental protocol, the stabilities/dynamics of these interactions may be better judged by the length of time that physical interactions are seen to persist in live imaging studies such as Chen et al. (2018), Vazquez et al. (2006) and Li et al. (2011).

The *in situ* FISH data we’ve seen also seems problematic in that probe hybridization results in a significant de-condensation of chromatin. For two probe sets complementary to adjacent ~1.2 kb DNA sequences, the measured center-to-center distance that we’ve seen was ~110 nM. This is about 1/3 the length that is expected for a 1.2 kb naked DNA fragment, and about 1.7 times larger than that expected for a beads-on-a-string nucleosome array (~60 nM). However, chromatin is thought to be compacted into a 30 nM fiber, which is estimated to reduce the length of DNA by at least another ~6 fold. If this estimate is correct, FISH hybridization would appear to result in a ~10 fold decompaction of chromatin. A decompaction of this magnitude would necessarily be followed by a significant distortion in the actual conformation of chromatin loops.

(4) The analysis of the Micro-C data appears to be largely qualitative. Key information about the number of reads sequenced, reaps mapped, and data quality are not presented. No quantitative framework for identifying features such as the "plumes" is described. The study and its findings would be strengthened by a more rigorous analysis of these rich datasets, including the use of systematic thresholds for calling patterns of organization in the data.

Additional information on the number of reads and data quality have been included in the Methods section.

(5) Related to Point 4, the lack of quantitative details about the Micro-C data make it difficult to evaluate if the changes observed are due to biological or technical factors. It is essential that the authors provide quantitative means of controlling for factors like sampling depth, normalization, and data quality between the samples.

The reviewer suggests that biological and/or technical differences between the four samples could account for the observed changes in the MicroC patterns for the *eve* TAD and its neighbors. If this were the case, then similar changes in MicroC patterns should be observed elsewhere in the genome. Since much of the genome is analyzed in these MicroC experiments, there is an abundance of internal controls for each experimental manipulation of the *nhomie* boundary. For two of the *nhomie* replacements, *nhomie reverse* and *homie forward*, the plume above the *eve* volcano triangle is replaced by clouds surrounding the *eve* volcano triangle. If these changes in the *eve* MicroC contact patterns are due to significant technical (or biological) factors, we should observe precisely the same sorts of changes in TADs elsewhere in the genome that are volcano triangles with plumes.

Author response image 9 shows the MicroC contact pattern for several genes in the Antennapedia complex. The *deformed* gene is included in a TAD which, like *eve*, is a volcano triangle topped by a plume. A comparison of the *deformed* MicroC contact patterns for *nhomie forward* (panel B) with the MicroC patterns for *nhomie reverse* (panel C) and *homie forward* (panel D) indicates that while there are clearly technical differences between the samples, these differences do not result in the conversion of the *deformed* plume into clouds as is observed for the *eve* TAD. The MicroC patterns elsewhere in Antennapedia complex are also very similar in all four samples. Likewise, comparisons of regions elsewhere in the fly genome indicate that the basic contact patterns are similar in all four samples. So while there are technical differences which are reflected in the relative pixel density in the TAD triangles and the LDC domains, these differences do not result in converting plumes into clouds nor do the alter the basic patterns of TAD triangles and LDC domains. As for biological differences—the embryos in each sample are at roughly the same developmental stage and were collected and processed using the same procedures. Thus, the biological factors that could reasonably be expected to impact the organization of specific TADs (e.g., cell-type specific differences) are not going to impact the patterns we see in our experiments.

**Author response image 9. sa3fig9:** 

(6) The ISH effects reported are modest, especially in the case of the HCR. The details provided for how the imaging data were acquired and analyzed are minimal, which makes evaluating them challenging. It would strengthen the study to provide much more detail about the acquisition and analysis and to include depiction of intermediates in the analysis process, e.g. the showing segmentation of stripes.

The imaging analysis presented in Fig. 5 is just standard confocal microscopy. Individual embryos were visualized and scored. An embryo in which stripes could be readily detected was scored as ‘positive’ while an embryo in which stripes couldn’t be detected was scored as ‘negative.’

**Recommendations for the authors:**

**Editor comments:**
It was noted that the Jaynes lab previously published extensive genetic evidence to support the stem loop and circle loop models of Homie-Nhomie interactions (Fujioka 2016 Plos Genetics) that were more convincing than the Micro-C data presented here in proof of their prior model. Maybe the authors could more clearly summarize their prior genetic results to further try to convince the reader about the validity of their model.
**Reviewer #1 (Recommendations For The Authors):**
Below, I list specific comments to further improve the manuscript for publication. Most importantly, I recommend the authors tone down their proposal that boundary pairing is a universal TAD forming mechanism.(1) The title is cryptic.

The title has been changed.

(2) The second sentence in the abstract is an overstatement: "In flies, TADs are formed by physical interactions between neighboring boundaries". Hi-C and Micro-C studies have not provided evidence that most TADs in *Drosophila* show focal interactions between their bracketing boundaries. The authors rely too strongly on prior studies that used artificial reporter transgenes to show that multimerized insulator protein binding sites or some endogenous fly boundaries can mediate boundary bypass, as evidence that endogenous boundaries pair.

Please see responses (1.1) and (1.3) and Author response images 1 and 7. Note that using dHS-C, most TADs that we’ve looked at so far are topped by a “dot” at their apex.

(3) Line 64: the references do not cite the stated "studies dating back to the '90's'".

The papers cited for that sentence are reviews which discussed the earlier findings. The relevant publications are cited at the appropriate places in the same paragraph.

(4) Line 93: "On the other hand, while boundaries have partner preferences, they are also promiscuous in their ability to establish functional interactions with other boundaries." It was unclear what is meant here.

Boundaries that (a) share binding sites for proteins that multimerize, (b) have binding sites for proteins that interact with each other, or (c) have binding sites for proteins that can be bridged by a third protein can potentially pair with each other. However, while these mechanisms enable promiscuous pairing interactions, they will also generate partner preferences (through a greater number of a, b, and/or c).

(5) It could be interesting to discuss the fact that it remains unclear whether Nhomie and Homie pair in cis or in trans, given that homologous chromosomes are paired in *Drosophila*.

The studies in Fujioka et al. (2016) show that *nhomie* and *homie* can pair both in *cis* and in *trans*. Given the results described in (1.2), we imagine that they are paired both in *cis* and in *trans* in our experiments.

(6) Line 321: Could the authors further explain why they think that "the nhomie reverse circle-loop also differs from the nhomie deletion (λ DNA) in that there is not such an obvious preference for which eve enhancers activate expression"?

The likely explanation is that the topology/folding of the altered TADs impacts the probability of interactions between the various *eve* enhancers and the promoters of the flanking genes. We have added a phrase to that sentence to make this more clear.

(7) The manuscript would benefit from shortening the long Discussion by avoiding repeating points described previously in the Results.(8) Line 495: "If, as seems likely, a significant fraction of the TADs genome-wide are circle loops, this would effectively exclude cohesin-based loop extrusion as a general mechanism for TAD formation in flies". The evidence provided in this manuscript appears insufficient to discard ample evidence from multiple laboratories that TADs form by compartmentalization or loop extrusion. Multiple laboratories have, for example, demonstrated that cohesin depletion disrupts a large fraction of mammalian TADs.

Points made here and in #9 have been responded to in (1.1), (2.1), and (2.4) above. We would suggest that the evidence for loop extrusion falls short of compelling (as it is based on the analysis of TAD neighborhoods, not TADs—that is forests, not trees) and, given the results reported in Goel et al. (in particular Fig. 4 and Sup. Fig. 8), is clearly suspect. This is not to mention the fact that cohesin loop extrusion can’t generate circle-loop TADs, yet circle-loops clearly exist. Likewise, as discussed in (2.4), it is not clear to us that the shared chromatin states, polymer co-segregation and co-repulsion, account for the compartmental patchwork patterns of TAD:TAD interactions. The results from the experimental manipulations in this paper and the accompanying paper, together with studies by others, e.g., Kyrchanova et al. (2008) and Mohana et al. (2023), would also seem to be at odds with the model for compartments as currently formulated.

The unique properties of Nhomie and Homie, namely the remarkable specificity with which they physically pair over large distances (Fujioka et al. 2016) may rather suggest that boundary pairing is a phenomenon restricted to special loci. Moreover, it has not yet been demonstrated that Nhomie or Homie are also able to pair with the TAD boundaries on their left or right, respectively.

Points made here were discussed in detail in (1.2). As described in detail in (1.2), it is not the case that *nhomie* and *homie* are “unique” or “special.” Other fly boundaries can do the same things. As for whether *nhomie* and *homie* pair with their neighbors: we haven’t done transgene experiments (e.g., testing by transvection or boundary bypass). Likewise, in MicroC experiments there are no obvious dots at the apex of the neighboring TADs that would correspond to *nhomie* pairing with the neighboring boundary to the left and *homie* pairing with the neighboring boundary to the right. However, this is to be expected. As we discussed in (1.3) above, only MNase-resistant elements will generate dots in standard MicroC experiments. On the other hand, when boundary:boundary interactions are analyzed by dHS-C (c.f., Author response image 8), there are dots at the apex of both neighboring TADs.

(9) The comment in point 8 also applies to the concluding 2 sentences (lines 519-524) of the Discussion.

See the response to #8 above. Otherwise, the concluding sentences are completely accurate. Validation of the cohesin loop-extrusion/CTCF roadblock model will required demonstrating (a) that all TADs are either stem-loops or unanchored loops, and (b) that TAD endpoints are always marked by CTCF.

The likely presence of circle-loops and the evidence of TAD boundaries that don’t have CTCF (c.f., Author response image 2 from Goel et al. 2023) already suggest that this model can’t (either fully or perhaps not even primarily) account for TAD formation in mammals.

(10) Figs. 3 and 6: It would be helpful to add the WT screenshot in the same figure, for direct comparison.

We think it is easy enough to scroll between figures, especially since *nhomie forward* looks just like WT.

(11) Fig. 6: It would be helpful to show a cartoon view of a circle loop to the right of the Micro-C screenshot, as was done in Fig. 3.

Good idea. This was added to the figure.

(12) Fig. 5: It would be helpful to standardize the labelling of the different genotypes throughout the figures and panels ("inverted" versus "reverse" versus an arrow indicating the direction).

This was fixed.

**Reviewer #2 (Recommendations For The Authors):**
Minor Points:(1) The Micro-C data does not appear to be deposited in an appropriate repository. It would be beneficial to the community to make these data available in this way.

This has been done.

(2) Readers not familiar with *Drosophila* development would benefit from a gentle introduction to the stages analyzed and some brief discussion on how the phenomenon of somatic homolog pairing might influence the study, if at all.

We included a rough description of the stages that were analyzed for both the *in situ*s and MicroC. We thought that a full description of what is going on at each of the stages wasn’t necessary, as the process of development is not a focus of this manuscript. In other studies, we’ve found that there are only minor differences in MicroC patterns between the blastoderm stage and stage 12-16 embryos. While these minor differences are clearly interesting, we didn’t discuss them in the text. In all of our experiments, chromosomes are likely to be paired. In NC14 embryos (the stage for visualizing *eve* stripes and the MicroC contact profiles in Fig. 2) replication of euchromatic sequences is thought to be quite rapid. While homolog pairing is incomplete at this stage, sister chromosomes are paired. In stage 12-16 embryos, homologs will be paired, and if the cells are arrested in G2, then sister chromosome will also be paired. So, in all of the experiments, chromosomes (sisters and/or homologs) are paired. However, since we don’t have examples of unpaired chromosomes, our experiments don’t provide any info on how chromosome pairing might impact MicroC/expression patterns.

(3) "P > 0.01" appears several times. I believe the authors mean to report "P < 0.01".

This was fixed.

References for Author Response to Reviewers

Batut PJ, Bing XY, Sisco Z, Raimundo J, Levo M, Levine MS. 2022. Genome organization controls transcriptional dynamics during development. Science. 375(6580):566-570.

Bing X, Ke W, Fujioka M, Kurbidaeva A, Levitt S, Levine M, Schedl P, Jaynes JB. 2024. Chromosome structure in *Drosophila* is determined by boundary pairing not loop extrusion. eLife2024;13:RP94114 DOI: https://doi.org/10.7554/eLife.94114.3

Blanton J, Gaszner M, Schedl P. 2003. Protein:Protein interactions and the pairing of boundary elements *in vivo*. Genes Dev. 17(5):664-675.

Bonchuk A, Boyko K, Fedotova A, Nikolaeva A, Lushchekina S, Khrustaleva A, Popov V, Georgiev P. 2021. Structural basis of diversity and homodimerization specificity of zinc-finger-associated domains in *Drosophila*. Nucleic Acids Res. 49(4):2375-2389.

Bonchuk A, Kamalyan S, Mariasina S, Boyko K, Popov V, Maksimenko O, Georgiev P. 2020. N-terminal domain of the architectural protein CTCF has similar structural organization and ability to self-association in bilaterian organisms. Sci Rep. 10(1):2677.

Chen H, Levo M, Barinov L, Fujioka M, Jaynes JB, Gregor T. 2018. Dynamic interplay between enhancer–promoter topology and gene activity. Nat Genet. 50(9):1296.

Fedotova AA, Bonchuk AN, Mogila VA, Georgiev PG. 2017. C2H2 zinc finger proteins: The largest but poorly explored family of higher eukaryotic transcription factors. Acta Naturae. 9(2):47-58.

Fujioka M, Ke W, Schedl P, Jaynes JB. 2024. The *homie* insulator has sub-elements with different insulating and long-range pairing properties. bioRxiv. 2024.02.01.578481.

Fujioka M, Mistry H, Schedl P, Jaynes JB. 2016. Determinants of chromosome architecture: Insulator pairing in *cis* and in *trans*. PLoS Genet. 12(2):e1005889.

Galloni M, Gyurkovics H, Schedl P, Karch F. 1993. The *bluetail* transposon: Evidence for independent cis‐regulatory domains and domain boundaries in the bithorax complex. The EMBO Journal. 12(3):1087-1097.

Gaszner M, Vazquez J, Schedl P. 1999. The Zw5 protein, a component of the *scs* chromatin domain boundary, is able to block enhancer-promoter interaction. Genes Dev. 13(16):2098-2107.

Goel VY, Huseyin MK, Hansen AS. 2023. Region capture micro-c reveals coalescence of enhancers and promoters into nested microcompartments. Nat Genet. 55(6):1048-1056.

Harris HL, Gu H, Olshansky M, Wang A, Farabella I, Eliaz Y, Kalluchi A, Krishna A, Jacobs M, Cauer G et al. 2023. Chromatin alternates between a and b compartments at kilobase scale for subgenic organization. Nat Commun. 14(1):3303.

Hart CM, Zhao K, Laemmli UK. 1997. The *scs'* boundary element: Characterization of boundary element-associated factors. Mol Cell Biol. 17(2):999-1009.

Hsieh TS, Cattoglio C, Slobodyanyuk E, Hansen AS, Rando OJ, Tjian R, Darzacq X. 2020. Resolving the 3D landscape of transcription-linked mammalian chromatin folding. Mol Cell. 78(3):539-553.e538.

Krietenstein N, Abraham S, Venev SV, Abdennur N, Gibcus J, Hsieh TS, Parsi KM, Yang L, Maehr R, Mirny LA et al. 2020. Ultrastructural details of mammalian chromosome architecture. Mol Cell. 78(3):554-565.e557.

Kyrchanova O, Chetverina D, Maksimenko O, Kullyev A, Georgiev P. 2008. Orientation-dependent interaction between *Drosophila* insulators is a property of this class of regulatory elements. Nucleic Acids Res. 36(22):7019-7028.

Kyrchanova O, Ibragimov A, Postika N, Georgiev P, Schedl P. 2023. Boundary bypass activity in the *Abdominal-B* region of the *Drosophila* bithorax complex is position-dependent and regulated. Open Biol. 13(8):230035.

Kyrchanova O, Kurbidaeva A, Sabirov M, Postika N, Wolle D, Aoki T, Maksimenko O, Mogila V, Schedl P, Georgiev P. 2018. The bithorax complex *iab-7* Polycomb response element has a novel role in the functioning of the *Fab-7* chromatin boundary. PLoS Genet. 14(8):e1007442.

Kyrchanova O, Sabirov M, Mogila V, Kurbidaeva A, Postika N, Maksimenko O, Schedl P, Georgiev P. 2019a. Complete reconstitution of bypass and blocking functions in a minimal artificial *Fab-7* insulator from *Drosophila* bithorax complex. Proceedings of the National Academy of Sciences.201907190.

Kyrchanova O, Wolle D, Sabirov M, Kurbidaeva A, Aoki T, Maksimenko O, Kyrchanova M, Georgiev P, Schedl P. 2019b. Distinct elements confer the blocking and bypass functions of the bithorax *Fab-8* boundary. Genetics.genetics. 302694.302019.

Kyrchanova O, Zolotarev N, Mogila V, Maksimenko O, Schedl P, Georgiev P. 2017. Architectural protein pita cooperates with dCTCF in organization of functional boundaries in the bithorax complex. Development. 144(14):2663-2672.

Li H-B, Muller M, Bahechar IA, Kyrchanova O, Ohno K, Georgiev P, Pirrotta V. 2011. Insulators, not Polycomb response elements, are required for long-range interactions between Polycomb targets in *Drosophila melanogaster*. Mol Cell Biol. 31(4):616-625.

Li X, Tang X, Bing X, Catalano C, Li T, Dolsten G, Wu C, Levine M. 2023. GAGA-associated factor fosters loop formation in the *Drosophila* genome. Mol Cell. 83(9):1519-1526.e1514.

Mohana G, Dorier J, Li X, Mouginot M, Smith RC, Malek H, Leleu M, Rodriguez D, Khadka J, Rosa P et al. 2023. Chromosome-level organization of the regulatory genome in the *Drosophila* nervous system. Cell. 186(18):3826-3844.e3826.

Muller M, Hagstrom K, Gyurkovics H, Pirrotta V, Schedl P. 1999. The *Mcp* element from the *Drosophila melanogaster* bithorax complex mediates long-distance regulatory interactions. Genetics. 153(3):1333-1356.

Muravyova E, Golovnin A, Gracheva E, Parshikov A, Belenkaya T, Pirrotta V, Georgiev P. 2001. Loss of insulator activity by paired *Su(hw)* chromatin insulators. Science. 291(5503):495-498.

Postika N, Metzler M, Affolter M, Müller M, Schedl P, Georgiev P, Kyrchanova O. 2018. Boundaries mediate long-distance interactions between enhancers and promoters in the *Drosophila* bithorax complex. PLoS Genet. 14(12):e1007702.

Samal B, Worcel A, Louis C, Schedl P. 1981. Chromatin structure of the histone genes of *D. melanogaster*. Cell. 23(2):401-409.

Sigrist CJ, Pirrotta V. 1997. Chromatin insulator elements block the silencing of a target gene by the *Drosophila* Polycomb response element (PRE) but allow trans interactions between PREs on different chromosomes. Genetics. 147(1):209-221.

Udvardy A, Schedl P. 1984. Chromatin organization of the 87A7 heat shock locus of *Drosophila melanogaster*. J Mol Biol. 172(4):385-403.

Vazquez J, Muller M, Pirrotta V, Sedat JW. 2006. The *Mcp* element mediates stable long-range chromosome-chromosome interactions in *Drosophila*. Molecular Biology of the Cell. 17(5):2158-2165.

Zolotarev N, Fedotova A, Kyrchanova O, Bonchuk A, Penin AA, Lando AS, Eliseeva IA, Kulakovskiy IV, Maksimenko O, Georgiev P. 2016. Architectural proteins Pita, Zw5,and Zipic contain homodimerization domain and support specific long-range interactions in *Drosophila*. Nucleic Acids Res. 44(15):7228-7241.